# CtBP impedes JNK- and Upd/STAT-driven cell fate misspecifications in regenerating *Drosophila* imaginal discs

**Melanie I Worley, Larissa A Alexander, Iswar K Hariharan\***

Department of Molecular and Cell Biology, University of California, Berkeley, Berkeley, United States

**Abstract** Regeneration following tissue damage often necessitates a mechanism for cellular reprogramming, so that surviving cells can give rise to all cell types originally found in the damaged tissue. This process, if unchecked, can also generate cell types that are inappropriate for a given location. We conducted a screen for genes that negatively regulate the frequency of notum-to-wing transformations following genetic ablation and regeneration of the wing pouch, from which we identified mutations in the transcriptional co-repressor *C-terminal Binding Protein (CtBP)*. When CtBP function is reduced, ablation of the pouch can activate the JNK/AP-1 and JAK/STAT pathways in the notum to destabilize cell fates. Ectopic expression of Wingless and Dilp8 precede the formation of the ectopic pouch, which is subsequently generated by recruitment of both anterior and posterior cells near the compartment boundary. Thus, CtBP stabilizes cell fates following damage by opposing the destabilizing effects of the JNK/AP-1 and JAK/STAT pathways.
DOI: https://doi.org/10.7554/eLife.30391.001

## Introduction

As development proceeds in multicellular organisms, cells become more restricted in developmental potential. This results initially from the expression of lineage-specific transcription factors and is then stabilized by heritable chromatin states that restrict accessibility of the transcriptional machinery to subsets of genes (*Britten and Davidson, 1969*; *Levine, 2010*; *Allis and Jenuwein, 2016*). The relative stability of these 'epigenetic landscapes' is thought to protect cells from patterns of gene expression that are inappropriate to that particular lineage. The paradigm that progressive restrictions in cell fate during development are unidirectional and irreversible was challenged by the demonstration that nuclei from differentiated cells could be reprogrammed to a more naïve state either by transferring them into an enucleated fertilized egg (*Gurdon, 1962*) or by expressing a combination of transcription factors (*Takahashi and Yamanaka, 2006*). Indeed, pluripotent stem cells have been derived from highly specialized cell types including neurons (*Kim et al., 2011*) and lymphocytes (*Loh et al., 2009*).

Even earlier, studies of *Drosophila* imaginal discs by Ernst Hadorn and colleagues showed that cells determined to one fate could switch to a very different fate. During embryogenesis, groups of cells at particular locations are specified to form particular imaginal discs (e.g. genital disc, wing disc, eye-antennal disc) depending on their location in the embryo (*Cohen, 1993*). Although these fates are determined during embryogenesis, disc cells do not differentiate to form adult structures until many days later, when metamorphosis occurs. To investigate the stability of the determined state, Hadorn's group transplanted imaginal disc fragments into abdomens of female adult flies where regeneration occurred. Most often, the tissue generated was appropriate for the implanted imaginal disc. However, at low frequency, regeneration resulted in tissue that was appropriate for a different disc (e.g. leg disc tissue generated from a fragment of genital disc). Hadorn termed this

**\*For correspondence:**
ikh@berkeley.edu

**Competing interests:** The authors declare that no competing interests exist.

**eLife digest** Some animals are more able to replace damaged tissue than others. A salamander, for example, can re-grow an amputated limb but a mouse or human cannot. After damage or injury certain types of cells are lost and need to be replaced by cells that are left behind. The remaining cells – or new cells that develop from them – must change their characteristics to better resemble the lost cells. This property, known as plasticity, needs to be controlled tightly. Excessive plasticity can result in forming tissues that are completely inappropriate for that location in the animal.

The fruit fly *Drosophila melanogaster* can be used to investigate plasticity during regeneration. Fruit fly larvae contain structures known as imaginal discs that can regenerate if damaged. Occasionally, when the imaginal discs regenerate, they produce the wrong kind of tissue.

Worley et al. set out to look for genes that would normally prevent such mistakes. Their search began with looking for flies with mutations that caused regeneration to go awry following damage. Specifically, Worley et al. looked for mutant flies that grew extra wings after a structure was damaged that would normally only generate a single wing. Once such flies had been found, further experiments were used to narrow down the search and confirm which gene was mutated.

This approach revealed that flies with mutations in the gene for a protein called CtBP (which is short for C-terminal binding protein) made more errors during regeneration and commonly regenerated inappropriate structures such as an extra wing. Importantly, mammals have very similar genes, but few researchers had previously studied if they also play a role in regeneration.

Worley et al. went on to show that CtBP dampens the activity of two signaling pathways (namely the JNK/AP-1 pathway and the JAK-STAT pathway), both of which promote plasticity. Thus, when CtBP levels are reduced, there is excessive plasticity.

These findings implicate CtBP as a regulator of plasticity during regeneration. This is an important first step in thinking of strategies that would allow researchers to guide and reshape the development of tissues during regeneration.

DOI: https://doi.org/10.7554/eLife.30391.002

phenomenon transdetermination (a switch from one determined state to another) (reviewed in [*Hadorn, 1968*; *McClure and Schubiger, 2007*; *Worley et al., 2012*]).

While transdetermination has been studied for years, many aspects are still mysterious. First, the fate change does not happen in a single cell, but rather, initiates in a small group of cells (*Gehring, 1967*; *Hadorn et al., 1970*; *Maves and Schubiger, 1998*). How a group of cells coordinately changes fate is not known. Second, certain fate changes seemed more likely than others. For example, genital-to-leg transformations or leg-to-wing transformations were far more frequent than transformations in the opposite direction. Third, transdetermination can be viewed as an aberrant form of regeneration that occurs following tissue damage and correlates with increased local proliferation (*Sustar and Schubiger, 2005*). Increased JNK activity, which is observed following damage (*Bosch et al., 2005*; *Mattila et al., 2005*; *Bergantiños et al., 2010*; *Fan et al., 2014*), reduces Polycomb-mediated repression and increases the frequency of transdetermination (*Lee et al., 2005*) as well as other types of fate changes (*Herrera and Morata, 2014*; *Schuster and Smith-Bolton, 2015*) by mechanisms that have not been elucidated. Finally, not all regions within a given disc have the same probability of undergoing a change of fate; Gerold Schubiger and colleagues described a region in the leg disc called the 'weak point' that has drastically higher rates of leg-to-wing transdetermination (*Schubiger, 1971*; *Maves and Schubiger, 1995*).

Our understanding of cell fate plasticity during regeneration is still at an early stage. Based on gene expression changes in regenerating discs, mutations in several candidate genes examined were shown to change the frequency of transdetermination, but the underlying mechanisms are not known (*Klebes et al., 2005*; *McClure et al., 2008*). To systematically screen for mutations that increase the frequency of cell fate changes during regeneration, we used a genetic system developed in our laboratory that reproducibly ablates a specific portion of the wing pouch and then allows it to regenerate (*Smith-Bolton et al., 2009*). We identified mutations in the gene encoding the transcriptional co-repressor, C-terminal Binding Protein (CtBP), and show that CtBP opposes the destabilization of cell fates during regeneration caused by JNK and the JAK/STAT pathway.

# Results

To study changes in cell determination during regeneration, we used a genetic system to damage specific parts of imaginal discs and investigated genetic alterations that modify the response of surviving cells. Damage was induced by expressing the pro-apoptotic gene, *eiger* (*UAS-egr*) under the control of the wing pouch driver, *rn-GAL4,* with temporal control provided by a temperature-sensitive version of the transcriptional repressor GAL80, *tub-GAL80ts* (**Figure 1A**). At 18°C for 7 days, development occurs normally to an early point in the third larval instar. Then, a temperature shift to 30°C for 40 hr inactivates GAL80ts allowing *UAS-egr* expression, leading to a large amount of

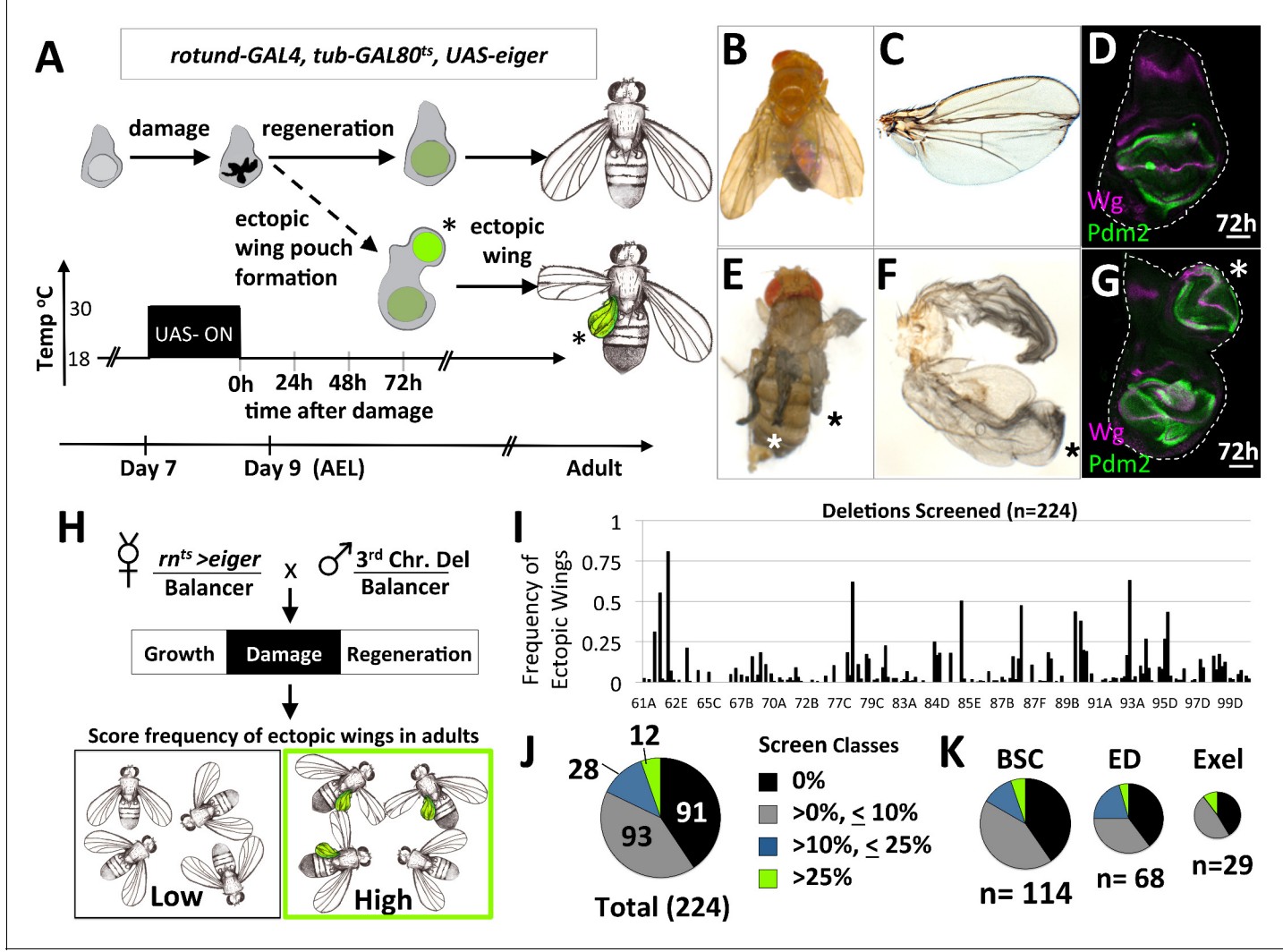

**Figure 1.** A system to study damage-induced cell-fate changes and a screen to identify genes that stabilize cell fates following damage. (A) The *rnts>eiger* (*rnts>egr*) genetic system is used to induce temporally-controlled, tissue ablation in the wing pouch of larval wing imaginal discs. (B–C) Regeneration following ablation of the pouch typically results in viable adults with wings of relatively normal appearance. (D) Wing discs after 72 hr of recovery show regeneration of the pouch, shown here stained with antibodies to pouch marker Pdm2 and Wg. (E–F) In some genetic backgrounds, ectopic wings are observed following ablation. (E) An adult with ectopic wings on both sides. (F) The wing and the ectopic wing detached from one side. (G) Wing disc with an ectopic pouch after 72 hr of recovery. Ectopic wings and ectopic pouches are indicated with an asterisk (*). (H) Schematic of the screen of deletions encompassing most of the third chromosome for enhancers of ectopic wing formation. Each deletion was tested by crossing to *rnts>egr*. Density-controlled cultures were developmentally synchronized by collecting L1 larvae, shifted on day 7 from 18°C to 30°C for 40 hr. Adults were scored for ectopic wings. (I) Graph of frequency of adults with one or more ectopic wings for each deletion screened, arranged by relative location in the third chromosome (cytogenetic positions indicated). (J, K) Deletion screen data binned into classes for the entire screen (J) and subsets of different deletion collections (K).

DOI: https://doi.org/10.7554/eLife.30391.003

apoptosis in the wing pouch. A return to 18°C enables functional GAL80$^{ts}$ to stop the expression of *UAS-egr* and allows regeneration to proceed (*Smith-Bolton et al., 2009*). This system will hereafter be referred to as *rn$^{ts}$>egr.*

Following damage induced using *rn$^{ts}$>egr*, most wing discs regenerated the lost tissue and the adults eclosed with wing blades of relatively normal size and a thorax of normal appearance (*Figure 1B,C*). Wing imaginal discs after 72 hr of recovery from damage had regenerated wing pouches (*Smith-Bolton et al., 2009*) (*Figure 1D*). However, in several genetic backgrounds, additional wing tissue was observed arising from the thorax posterior to the normal wing, in the region of the scutellum, at frequencies approaching as high as 40% of adults (*Figure 1E,F*). Normal halteres were still observed, indicating that the ectopic wings were not generated by a haltere-to-wing transformation (such as those caused by mutations in the Hox gene *Ubx*). In these same genotypes, wing discs often had a second (ectopic) wing pouch protruding from the posterior side of the notum (*Figure 1G*).

## A screen for mutations that increase cell fate plasticity following tissue damage identifies CtBP

The increased frequency of ectopic wings in some genetic backgrounds suggests that there are genes whose normal function includes the stabilization of cell fates during damage and regeneration. To identify such genes, we systematically screened 226 stocks bearing deletions of the third chromosome for their ability, when heterozygous, to increase the frequency of ectopic wings following *egr*-mediated damage and regeneration (*Figure 1H*). The frequency of adults with ectopic wings (EW) was scored for each deletion and graphed based on relative position of the deletion in the genome (*Figure 1I*). Taken together, these deletions covered approximately 90% of the euchromatic portion of chromosome 3. Of these 226 deletions tested, two were lethal, 91 elicited no EWs, 93 elicited the formation of EWs at a low penetrance (≤10%), 28 showed an intermediate penetrance (>10%,<25%) and 12 showed a high frequency of EWs (>25%)(*Figure 1J*). Deletions from different deletion collections, Bloomington Stock Center (BSC), DrosDel Project (ED) and Exelixis (Exel) each had a similar breakdown of classes (*Figure 1K*). The score for each deletion tested is in (*Supplementary file 1*).

We focused on the 87D region, since multiple overlapping deletions allowed us to narrow the region of interest to a small number of genes. We then tested the available alleles and identified the responsible gene as *C-terminal Binding Protein (CtBP)*. A small deletion (*Df(3R)Exel8157*) that spans *CtBP* caused EWs in 47% of adults (*Figure 2A*). A hypomorphic *CtBP* allele, *CtBP$^{03463}$*, and a loss-of-function allele, *CtBP$^{87De-10}$* (which we will henceforth refer to by its protein truncating mutation *CtBP$^{Q229*}$*) caused EWs in 22% and 42% of adults, respectively (*Figure 2A*). Mock ablation, with only either 1) *rn-GAL4, tub-GAL80$^{ts}$* or 2) *tub-GAL80$^{ts}$, UAS-egr* did not yield EWs (*Figure 2A*). EWs were attached to the scutellum and were often unable to flatten and appeared round (*Figure 2B*). The highest frequency of EWs occurred when the genetic damage was initiated on day seven after egg lay (AEL) in *CtBP$^{Q229*/+}$* mutants, while no EWs were observed in wild type on all days tested (*Figure 2—figure supplement 1*).

CtBP was originally identified as a protein that interacted with the C-terminus of the adenovirus E1A protein (*Boyd et al., 1993*). It has subsequently been shown to function as a transcriptional co-repressor in mammals and in *Drosophila* (*Nibu et al., 1998*; *Zhang and Levine, 1999*). While mammals have two paralogs of *CtBP,* reviewed in (*Chinnadurai, 2003*), the *Drosophila* genome contains a single *CtBP* gene. CtBP itself does not bind DNA. Rather, it associates with multiple DNA-binding proteins that possess a PxDLS motif, or related motif, including Hairy (*Poortinga et al., 1998*), Krüppel (*Nibu et al., 1998*) and Brinker (*Hasson et al., 2001*). The *Drosophila CtBP* gene has multiple spliced isoforms that are predicted to generate seven different proteins, which are usually grouped as short (379–386 aa) or long (473–481 aa) (*Figure 2C*).

To confirm that the increased frequency of EWs was in fact due to the disruption of *CtBP*, we used CRISPR/Cas9 to generate *CtBP* alleles on an isogenized Oregon-R third chromosome that did not cause EWs following damage (0%, n = 1892 adults) (*Figure 2D*). We generated two CRISPR guide RNAs that target regions spanning codons shared between all predicted protein isoforms, located at codon 148 and 334 (*Figure 2C*). The alleles recovered include a likely hypomorph, *CtBP$^{N148PY}$* (three base pair insertion), which induced EWs in approximately 11% of adults and a suspected null, *CtBP$^{148Δ2}$* (frame shift inducing two base pair deletion), which induced EWs in

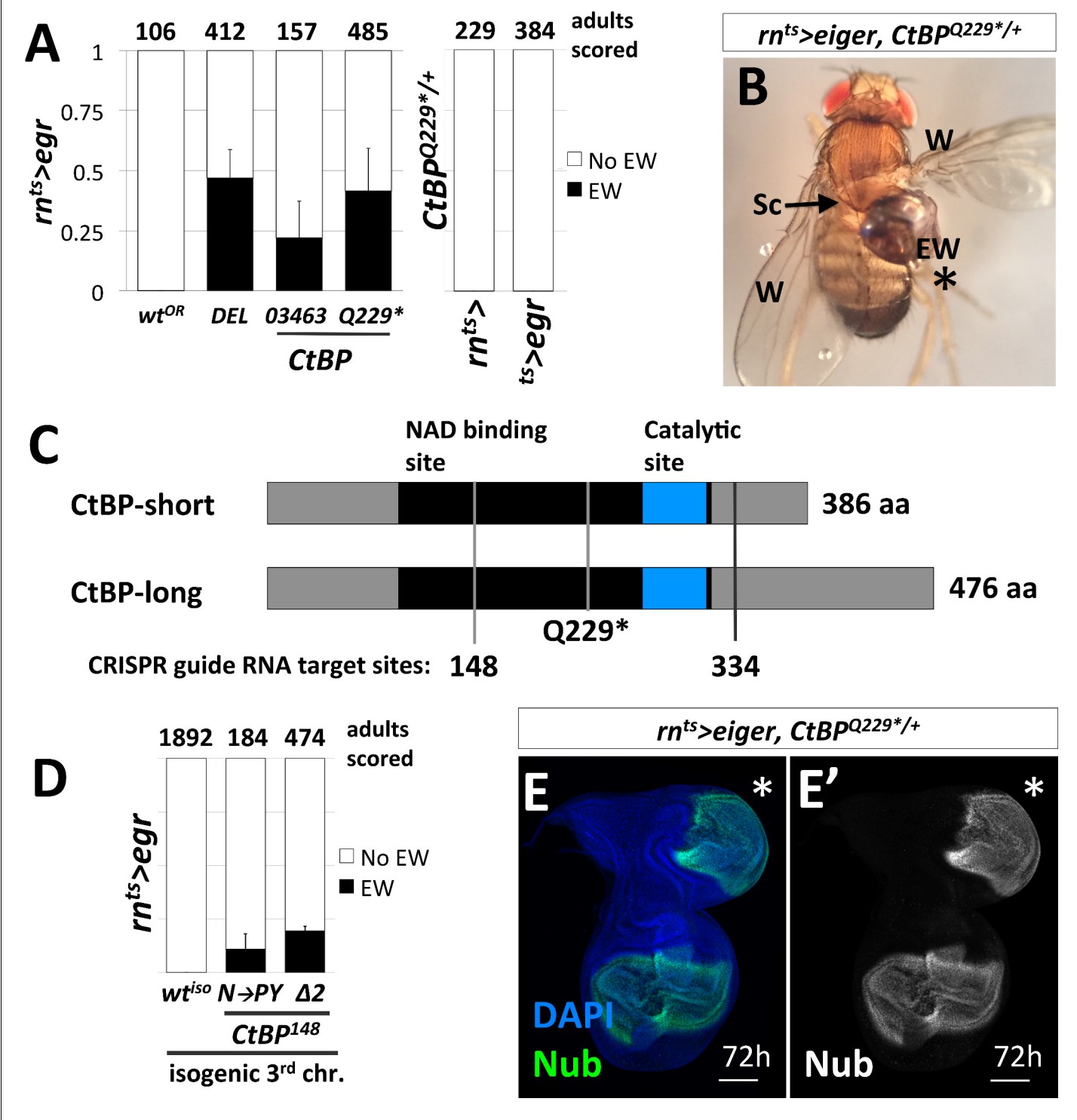

**Figure 2.** Loss-of-function mutations in *C-terminal Binding Protein* (*CtBP*) enhance the frequency of damage-induced ectopic wing formation. (A) Frequency of ectopic wings (EW) following *rn^ts^>egr* damage or mock ablation. EW scored in heterozygous genetic backgrounds for a wild-type Oregon R strain (*wt^OR^*), a deletion that spans *CtBP* (*DEL = Df(3R)Exel8157*), and two previously characterized alleles of *CtBP*, *CtBP^03463^* and *CtBP^Q229*^*. Mock ablation controls 1) *rn-GAL4, tub-GAL80^ts^* (*rn^ts^>*) and 2) *tub-GAL80^ts^, UAS-egr* (*^ts^>egr*), conducted in *CtBP^Q229*/+^* genetic background. (B) An ectopic wing in *CtBP^Q229*/+^* following ablation and regeneration. EW = ectopic wing, Sc = scutellum, W = wing. (C) A schematic of the two major protein isoforms of CtBP (short and long). The amino acid residues that correspond to codons at the CRISPR guide RNA target sites are indicated (148 and 334). (D) Frequency of EW following *rn^ts^>egr* damage in a wild type isogeneic background (*wt^iso^*) and CRISPR-generated *CtBP* alleles (*CtBP^N148PY^* (N–

*Figure 2 continued on next page*

*Figure 2 continued*

to–PY), CtBP$^{148\Delta2}$ (Δ2)) in that background. (E) CtBP$^{Q229*/+}$ wing imaginal disc following 72 hr of recovery stained with an antibody to Nubbin (Nub), a wing pouch marker (**Ng et al., 1995**). The ectopic pouch is marked with an asterisk (*). All bar graphs show averages between multiple experiments, error bars show standard deviations, and the numbers listed above the bars are the total adults scored.
DOI: https://doi.org/10.7554/eLife.30391.004
The following figure supplement is available for figure 2:

**Figure supplement 1.** Ectopic wings occur most frequently when induced on day seven in CtBP$^{Q229*/+}$.
DOI: https://doi.org/10.7554/eLife.30391.005

approximately 20% of adults (**Figure 2D**). When compared to the isogenic parent chromosome, CtBP$^{-/+}$ increases the frequency of EWs. Ectopic pouches were observed in CtBP$^{-/+}$ wing discs following damage and recovery (**Figure 2E**).

## Early molecular events in the formation of the ectopic pouch

To investigate what was triggering the formation of EWs, we focused on the early stages of ectopic pouch formation. Wingless (Wg) plays a key role in specifying the wing pouch in early development and is also upregulated during regeneration (**Smith-Bolton et al., 2009**). We compared Wg expression and apoptosis following damage in wild type (**Figure 3A–C**) and CtBP$^{Q229*/+}$ (**Figure 3D–F**) discs. At 0 hr of recovery, both wild type and CtBP$^{Q229*/+}$ discs had apoptotic cells and cellular debris (which stain for DCP-1) and high Wg expression around the damaged wing pouch. In the wild type discs, the normal notum Wg stripe was maintained (**Figure 3A'–C'**). In contrast, in ~45% of CtBP$^{Q229*/+}$ discs (n > 50), there was a spot of increased Wg expression along the posterior edge of the notum, along with a disruption of the notum Wg stripe (**Figure 3D'**). Over the next 24 hr to 48 hr of recovery and regeneration, ~40% of CtBP$^{Q229*/+}$ wing discs showed a domain of Wg expression accompanying morphological changes associated with the outgrowth of a wing pouch from the posterior edge of the notum (**Figure 3E,F**). Note that apoptosis was also detected in this region after 24 hr and 48 hr of recovery (**Figure 3E', F'**).

During regeneration, wg expression is upregulated through a damage-dependent wg enhancer, which is JNK-dependent (**Schubiger et al., 2010**; **Harris et al., 2016**). We found that this damage-dependent wg enhancer (BRV-B-GFP) drives reporter expression at the area of fate change in a subset of CtBP$^{Q229*/+}$ discs, but not in wild type (**Harris et al., 2016**), following rn$^{ts}$>egr damage (**Figure 3G,H**). In addition, the removal of one copy of the damage-dependent wg enhancer, which is deleted in the wg$^1$ allele (**Schubiger et al., 2010**), reduced the frequency of ectopic wings (**Figure 3I**). This is consistent with the ectopic pouch forming in a similar manner to the regenerating wing pouch (**Harris et al., 2016**).

It is known that ectopic wing pouches can be generated by the constitutive expression of UAS-wg along the anteroposterior compartment boundary in the absence of damage (**Ng et al., 1996**). We determined that ptc$^{ts}$>wg expression initiated during the second instar triggered ectopic wings, but expression initiated during the third instar did not for both wild type and CtBP$^{Q229*/+}$ (**Figure 3—figure supplement 1**). This is in contrast to the fate change triggered by rn$^{ts}$>egr in the third instar and suggests that Wg alone is insufficient to reprogram cell fates as the disc matures. In agreement with this finding, the overexpression of rn$^{ts}$>wg alone, or co-expression of rn$^{ts}$>egr and >wg did not enhance the frequency of ectopic wings (**Figure 3J**). Together, this suggests that other signaling pathways are critical for triggering ectopic wings at this point in development.

Since the BRV-B wg enhancer is activated by the JNK pathway, we examined other indicators of increased JNK signaling: AP-1-RFP (**Chatterjee and Bohmann, 2012**) and Matrix Metalloprotease 1 (MMP1) protein (**Page-McCaw et al., 2003**) expression. At 0 hr of recovery, both AP-1-RFP and MMP1 were expressed robustly around the egr-ablated wing pouch, and in many CtBP$^{Q229*/+}$ discs, at a second location in the notum (**Figure 3—figure supplement 2**). AP-1-RFP activity was observed in an ectopic pouch at 48 hr of recovery (**Figure 3—figure supplement 2**), suggesting that JNK activity at the secondary site may be important for triggering ectopic wing pouch formation in CtBP$^{-/+}$ mutants.

Previous studies found that dilp8 was highly upregulated during leg-to-wing transdetermination (**Klebes et al., 2005**) and also in regenerating imaginal discs (**Katsuyama et al., 2015**;

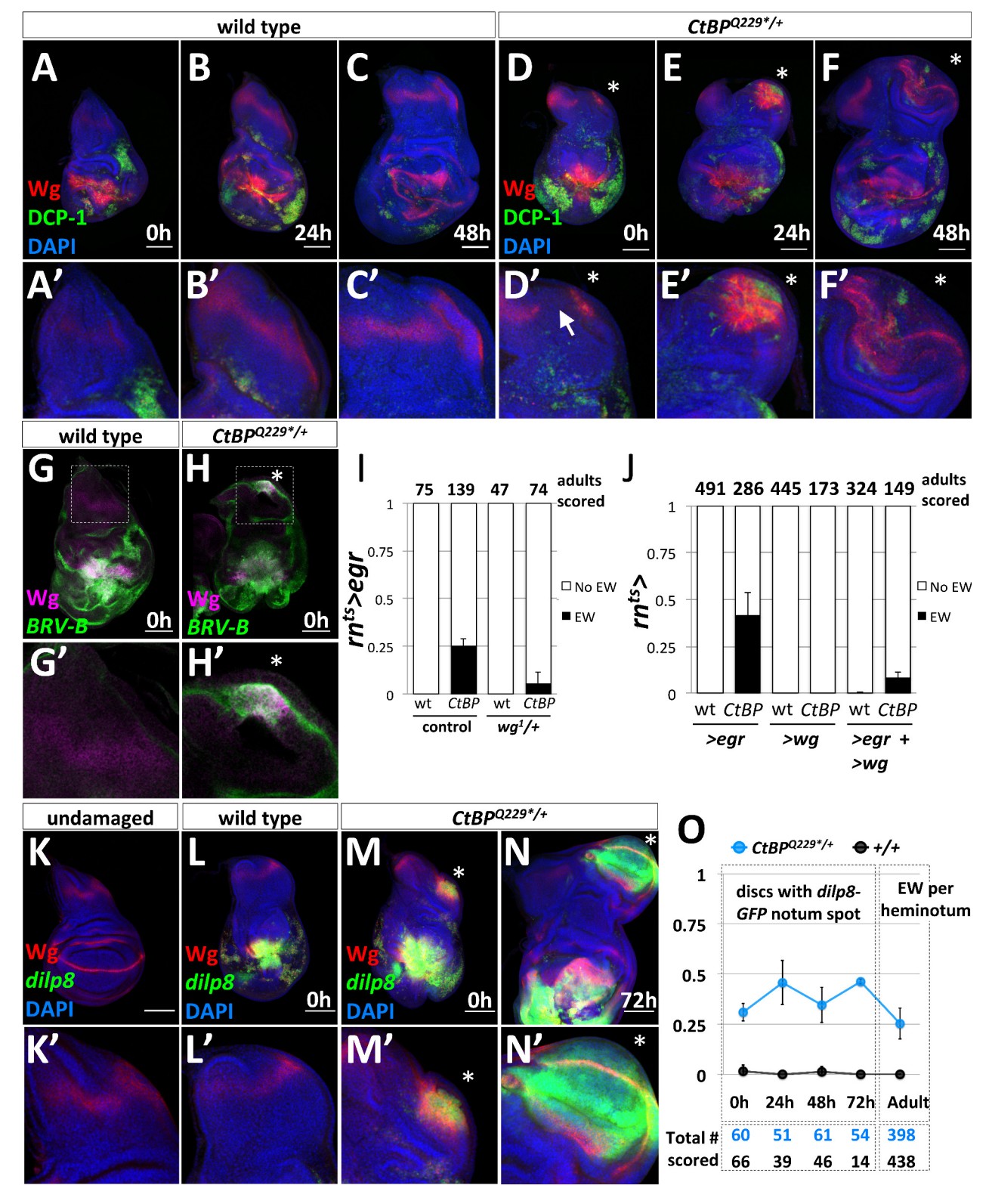

**Figure 3.** The formation of ectopic pouches is marked by the activation of Wg and JNK at a secondary location. (A–F) Wg and apoptosis visualized in wing discs following *rn^{ts}>egr* ablation of the pouch at different time points during recovery and regeneration (0 hr, 24 hr, and 48 hr after the downshift) for wild type (A–C) and CtBP^{Q229*/+} (D–F). Apoptotic cells and their debris stain with an antibody to cleaved Drosophila caspase-1 (DCP-1). (A'–F') Higher magnification images of the notum region. The site of secondary pouch formation is indicated with an asterisk (*). The arrow in (D') points to a

*Figure 3 continued on next page*

Figure 3 continued

region where the normal stripe of Wg expression in the notum is disrupted. (G–H) Damage-dependent *wg* enhancer, *BRV-B-GFP,* following *rn^ts^>egr* damage at 0 hr recovery in wild type (G) and CtBP^Q229*/+^ (H) Boxed regions are shown at higher magnification in (G'–H'). (I) Frequency of ectopic wings (EW) in adults following *rn^ts^>egr* damage with two (control) or one copy of the damage-dependent *wg* enhancer (*wg^1/+^*), in wild type and CtBP^Q229*/+^. (J) Frequency of EW in adults following overexpression of *UAS-egr* or *UAS-wg* alone, or *UAS-egr* and *UAS-wg* together driven by *rn^ts^>* in wild type and CtBP^Q229*/+^. (K) Expression of *dilp8-GFP* (*dilp8^MI00727^*) in an undamaged disc. (L–N) Expression of *dilp8-GFP* in damaged discs after 0 hr of recovery for wild type (L) and CtBP^Q229*/+^ (M) and after 72 hr of recovery for CtBP^Q229*/+^ (N). (O) Graph of frequency of discs with a spot of *dilp8-GFP* expression at different time points of recovery (0 hr, 24 hr, 48 hr, 72 hr), along with the frequency of ectopic wings per side (or heminotum) in adults.

DOI: https://doi.org/10.7554/eLife.30391.006

The following figure supplements are available for figure 3:

**Figure supplement 1.** *wingless* overexpression is not sufficient to induce ectopic wings on day 7.
DOI: https://doi.org/10.7554/eLife.30391.007
**Figure supplement 2.** JNK activity at secondary location in the notum and in the ectopic pouch.
DOI: https://doi.org/10.7554/eLife.30391.008
**Figure supplement 3.** Timing of *dilp8-GFP* spot appearance in the notum relative to Tsh downregulation and onset of Nub expression.
DOI: https://doi.org/10.7554/eLife.30391.009
**Figure supplement 4.** Bilateral ectopic wings slightly more frequent than expected for completely independent events.
DOI: https://doi.org/10.7554/eLife.30391.010
**Figure supplement 5.** *dilp8* is needed for EW in adults and additional *dilp8* increases the frequency.
DOI: https://doi.org/10.7554/eLife.30391.011

*Skinner et al., 2015*; *Harris et al., 2016*). *dilp8* encodes a secreted protein that is generated in response to abnormalities in tissue growth, likely in response to increased AP-1 activity, and Dilp8 expression delays pupariation (*Colombani et al., 2012*; *Garelli et al., 2012*; *Colombani et al., 2015*; *Garelli et al., 2015*; *Vallejo et al., 2015*; *Jaszczak et al., 2016*). A Minos insertion in the *dilp8* locus, *dilp8^MI00727^*, functions as a transcriptional reporter of *dilp8* via the expression of *GFP*, which we will refer to as *dilp8-GFP.* Under normal undamaged conditions, there was very low expression of *dilp8-GFP* in the wing disc (*Figure 3K*). As expected, damage with *rn^ts^>egr* resulted in high *dilp8-GFP* expression around the damaged and regenerating wing pouch (*Figure 3L*). In a subset of CtBP^Q229*/+^ discs there was a second spot of *dilp8-GFP* expression in the notum at 0 hr of recovery (*Figure 3M*), which largely overlapped with the area of increased Wg expression. After 72 hr of recovery, *dilp8-GFP* was still detectable in a morphologically distinct outgrowth, which by this stage expressed Wg in the characteristic pattern of a mature wing pouch (*Figure 3N*). At 0 hr of recovery, *dilp8-GFP* expression in the notum overlapped with MMP1 expression (*Figure 3—figure supplement 2*), which is consistent with JNK activity contributing to *dilp8-GFP* expression. *dilp8-GFP* in the notum was expressed in cells that still expressed the hinge and notum marker Teashirt (Tsh) and before the pouch marker Nubbin (Nub) was detected (*Figure 3—figure supplement 3*) suggesting that it may be an early marker of an impending fate change in notum cells.

To determine if early *dilp8-GFP* expression in the notum was predictive of ectopic pouch formation, we quantified its frequency at different recovery time points in wild type and CtBP^Q229*/+^ discs and the frequency of EWs per side in adults (*Figure 3O*). The frequency of secondary *dilp8-GFP* spots was relatively constant across time points in CtBP^Q229*/+^ discs and absent in wild-type discs. In addition, the frequency of EW per adult heminotum roughly matched the frequency of *dilp8-GFP* spots in discs. Therefore, in this system, the secondary spot of *dilp8-GFP* expression in the notum is likely an early predictor of future ectopic pouch and ectopic wing blade formation.

Do the events that trigger ectopic wings occur independently between the two wing imaginal discs within one larva? We recorded the number of CtBP^Q229*/+^ adults with ectopic wings absent (63%), on one side (23%) and on both sides (14%) (*Figure 3—figure supplement 4*). Slightly more adults had bilateral ectopic wings than predicted for completely independent events. This could be due to identical timing of the discs or long-range signals (circulating factors) that simultaneously promote, or are permissive for, fate changes in both discs.

In support of the idea of possible circulating factors promoting a permissive environment for ectopic pouch formation, we found that as animals homozygous for the loss-of-function allele, *dilp8^MI00727^*, do not regenerate well (as seen in [*Skinner et al., 2015*]) and also lacked ectopic wings (EW) (*Figure 3—figure supplement 5*). Conversely, we found that the addition of *UAS-dilp8*

enhanced the frequency of EWs, without triggering EWs on its own (*Figure 3—figure supplement 5*). Together this suggests that a developmental delay is important for EW formation.

## The ectopic wing pouch arises from cells in the notum

To determine if the ectopic pouch arises from a single cell changing its fate or from multiple cells, we generated marked clones of three varieties by *hsFLP*-induced recombination during the damage period and observed the clones following the growth of the ectopic pouch. The marked clones were arranged radially in the ectopic pouch yet only occupied a small fraction of it (estimated to be 10%) (*Figure 4A*). Thus, the ectopic pouch was generated either from multiple cells simultaneously changing fate or by the progressive recruitment of additional cells.

To determine if the cells that give rise to the ectopic pouch had originated in the notum, we lineage-marked cells using the lexA system. This allowed us to use the GAL4/UAS system to ablate the pouch and to permanently label cells that expressed the lexA lines with *lexAOp-FLP, Ubi<stop<GFP^{nls}*. *R76B02-lexA* was expressed primarily in the notum in the absence of damage (*Figure 4—figure supplement 1*). Following regeneration, a large number of notum cells were GFP labeled in the control discs (*Figure 4B*) and a significant proportion of the cells in the ectopic pouch was GFP labeled in *CtBP^{-/+}* discs (*Figure 4C*). Therefore, the cells that gave rise to the ectopic pouch once expressed *R76B02-lexA,* which indicates that these cells were once notum tissue, but had undergone a fate change to generate the ectopic pouch. Labeling with two additional *lexA* lines also supported the conclusion that a fate change occurred from notum to ectopic pouch (*Figure 4—figure supplement 1*).

A key transcription factor in the development of the notum is Mirror (Mirr) (*Diez del Corral et al., 1999*). The boundary between the notum and the hinge is located at a prominent fold at the edge of the *mirr-lacZ* expression domain (*Wang et al., 2016*) (*Figure 4D*). We observed the ectopic Wg expression in the notum within the *mirr-lacZ* expression domain (*Figure 4E*), and evidence of a new notum/hinge boundary in the ectopic growth (*Figure 4F*). In addition, removing one copy of *mirr* increased the frequency of ectopic wings in both wild type and *CtBP* mutant backgrounds (*Figure 4G*), suggesting that reducing the expression of notum-specifying factors increases the probability of damage-induced fate change, especially when *CtBP* levels are also reduced.

## The ectopic pouch is derived from both anterior and posterior cells

The ectopic pouch was usually composed of both anterior (A) and posterior (P) cells, yet the initial events appeared to occur near the posterior edge of the disc, raising the possibility that posterior cells might have changed to an anterior identity. Indeed, when we examined *dilp8-GFP* expression in discs following ablation, with the A and P compartments visualized, we found that the entire cluster of *dilp8-GFP* expressing cells was usually located posterior to the compartment boundary (*Figure 4H*). For discs at 0 hr of recovery with *dilp8-GFP* or *AP-1-RFP* expression in the notum, the expressing cells were scored as P in 17/20 discs, as A in 2/20 discs and as both A and P in 1/20 discs. Then, during the growth of the ectopic pouch, *dilp8-GFP* expression was observed in both A and P cells (*Figure 4I,J*). When we labeled individual cells two days prior to the ablation and examined the clones that they generated, we found that even large clones in the ectopic pouch did not cross the compartment boundary (n = 7) (*Figure 4K*). Therefore, the ectopic pouch was likely generated from both A and P precursor cells.

Taken together, these experiments show that the ectopic pouch arose from a group of cells in the notum that is composed of both A and P cells. These cells collectively underwent a notum-to-pouch fate change while preserving their compartmental identities. The appearance of the *dilp8* expression first in the P compartment and later in the A compartment suggests that the initial fate change recruited cells from across the compartment boundary to generate the ectopic pouch.

## Re-specifying cell fates requires the activation of the JNK/AP-1 pathway

We have shown that JNK signaling was increased in the notum at an early stage in the development of an ectopic pouch. Moreover, our system is based on the targeted expression of Egr, which is the *Drosophila* Tumor-Necrosis Factor-α (TNF-α) ligand that activates JNK-dependent apoptosis (*Igaki et al., 2002*) in the wing pouch. Thus JNK-signaling may function in multiple ways to promote

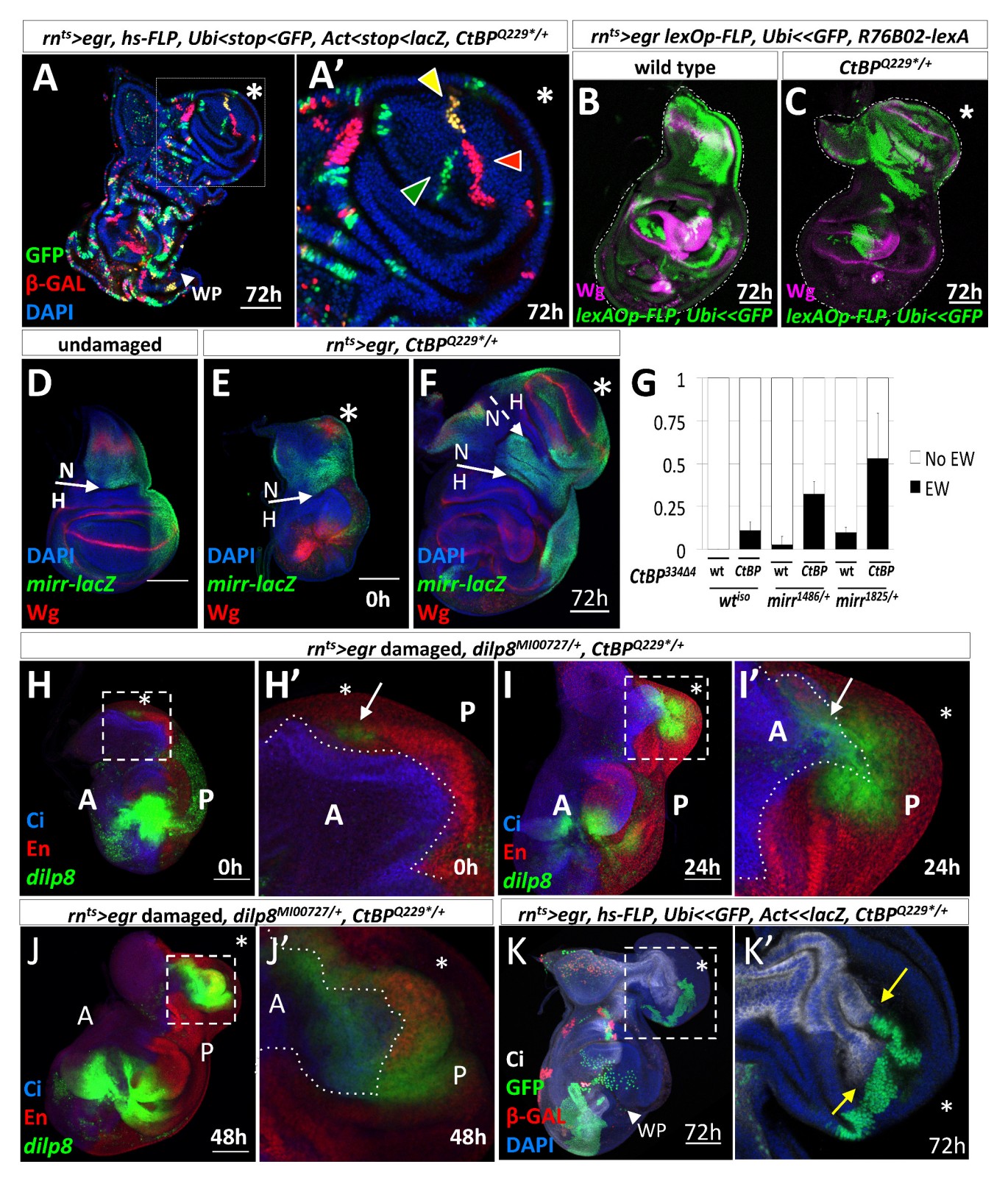

**Figure 4.** Characterization of the origin of cells that contribute to the ectopic pouch. (A) $rn^{ts}>egr$, $CtBP^{Q229*/+}$ wing disc with randomly-generated clones (two independent FLP-out constructs, $Act<stop<lacZ^{nls}$ and $Ubi<stop< GFP^{nls}$) generated with a 10 min heat-shock (37°C) at the start of the temperature shift from 18° C to 30° C and dissected 72 hr after the downshift. In (A'), there are three uniquely labeled clones (only $Ubi$-GFP, only $Act$-$lacZ$, or both), as indicated by arrowheads of matching color, in the ectopic pouch. (B–C) Lineage tracing following $rn^{ts}>egr$ induced damage in wild

*Figure 4 continued on next page*

*Figure 4 continued*

type (**B**) and *CtBP^Q229*/+* (**C**) genetic backgrounds with *R76B02-lexA, lexAOp-FLP, Ubi<stop<GFP^nls* after 72 hr of recovery. (**D–F**) *rn^ts>egr, CtBP^Q229*/+* wing discs with the enhancer trap *mirr-lacZ* stained with anti-Wg. (**D**) Undamaged. (**E–F**) Damaged discs, after 0 hr (**E**) and 72 hr (**F**) of recovery. N indicates notum, H indicates hinge. The asterisk indicates area of increased Wg expression in (**E**) and the ectopic wing pouch in (**F**). (**G**) Frequency of ectopic wings (EW) following *rn^ts>egr* damage in wild type or *CtBP^334Δ4/+* together with different *mirr* alleles, *mirr^1486/+* and *mirr^1825/+*. Error bars show standard deviations. (**H–J**) *rn^ts>egr, CtBP^Q229*/+*wing discs at different time points of recovery with *dilp8-GFP*, anti-Cubitus Interruptus (Ci) for marking the anterior compartment and anti-Engrailed (En) for marking the posterior compartment. Note that *dilp8-GFP* expression initiates in the posterior compartment and then spreads to the anterior compartment. (**K**) *rn^ts>egr, CtBP^Q229*/+* wing discs containing randomly generated marked clones (similar to **A**) that were induced on day 5 (48 hr before the temperature shift). Note the large GFP-positive clone in the posterior compartment of the ectopic pouch that abuts a significant portion of the anterior/posterior compartment boundary. Dotted boxes indicate areas that are enlarged in subsequent panels. The time of recovery (i.e. after downshift) is indicated in the bottom right corner of the relevant panel.

DOI: https://doi.org/10.7554/eLife.30391.012

The following figure supplement is available for figure 4:

**Figure supplement 1.** Lineage tracing of cells from specific regions of the notum using *lexA* lines.

DOI: https://doi.org/10.7554/eLife.30391.013

the notum-to-pouch fate change. To test if JNK signaling was required, we reduced JNK signaling either throughout the disc or only within the GAL4-expressing domain. The removal of one copy of the gene encoding the JNK kinase, *hemipterous* (*hep),* reduced the frequency of ectopic wings (*Figure 5A*). Blocking JNK signaling autonomously in the *rn-GAL4* domain with the expression of a dominant negative form of JNK *(>JNK^DN)* or of the AP-1 transcription factor Fos (*>Fos^DN*) completely suppressed the formation of ectopic wings (*Figure 5A*), and also reduced the ablation of the wing pouch (*Figure 5—figure supplement 1*). Thus, JNK/AP-1 signaling was needed for the generation of ectopic wings, either through JNK-mediated apoptosis or other JNK-mediated signaling events.

We found, however, that ablation of the pouch alone was not sufficient to produce ectopic wings, as the expression of the pro-apoptotic gene *reaper* (*rpr*) did not produce ectopic wings, even in a *CtBP^Q229*/+* background (*Figure 5B*). Although *rn^ts>rpr* leads to a more effective ablation of the wing pouch than *rn^ts>egr,* it does not activate JNK to the same levels (*Smith-Bolton et al., 2009*; *Harris et al., 2016*). We tested if cell-autonomous methods of increasing JNK activity would lead to ectopic wings by expressing either a wild-type or constitutively active form of *hep* (*rn^ts>hep^wt,* *rn^ts>hep^CA*) (*Figure 5B*). Ectopic wings were only observed at very low frequency with *rn^ts>hep^CA*, which could conceivably be the result of cell non-autonomous JNK activation via Egr expression (*Pérez-Garijo et al., 2013*). Reducing the amount of apoptosis by shortening the ablation period from 40 hr to 24 hr reduced the frequency of EW for *rn^ts>egr* from ~45% to ~11% in *CtBP^Q229*/+*, and did not produce any EWs for either *rn^ts>rpr* or *rn^ts>hep^CA* (*Figure 5—figure supplement 2*). Egr signals through the receptor Grindelwald (Grind) to activate JNK signaling (*Andersen et al., 2015*). Overexpression of the intracellular domain of Grind (*rn^ts>grind^ICD*) induced apoptosis, yet did not produce EWs (*Figure 5B*). However, expression of a secreted form of Egr (*rn^ts>ecto*-egr) (*Narasimamurthy et al., 2009*), caused EWs at a slightly higher frequency than the full-length (membrane-tethered) form (*Figure 5B*). Thus, while the ablation of the pouch by Egr was necessary for the formation of EWs, some local spread of Egr also appeared necessary.

In order to determine where *egr* must be expressed to trigger ectopic pouch formation, we closely examined the expression domain of *rn-GAL4* with G-TRACE and detected expression in the wing pouch, a subset of the myoblasts and, at low levels, in scattered notum cells (*Figure 5—figure supplement 3*). Experiments with other GAL4 drivers determined that the expression of *egr* in the pouch was critical to the formation of ectopic wings and that the expression of *egr* in the notum or myoblasts was not sufficient (*Figure 5—figure supplement 3*).

Egr signaling activates both JNK signaling and apoptosis. While it is difficult to separate high-JNK activity from apoptosis, reducing apoptosis either by removing one copy of the pro-apoptotic genes *rpr, hid,* and *grim* (deleted in *Df(3L)XR38*), or by co-expressing a dominant-negative form of an effector caspase (*>Dronc^DN*) did not affect EW frequency (*Figure 5—figure supplement 4*). Co-expression of the anti-apoptotic gene *UAS-p35* with *rn^ts>egr*, still generated EWs in the adults (*Figure 5—figure supplement 4*). Conversely, increasing apoptosis by co-expression of *rn^ts>ecto*-egr and *rn^ts>rpr* suppressed the formation of ectopic wings. This suggests that a population of cells that

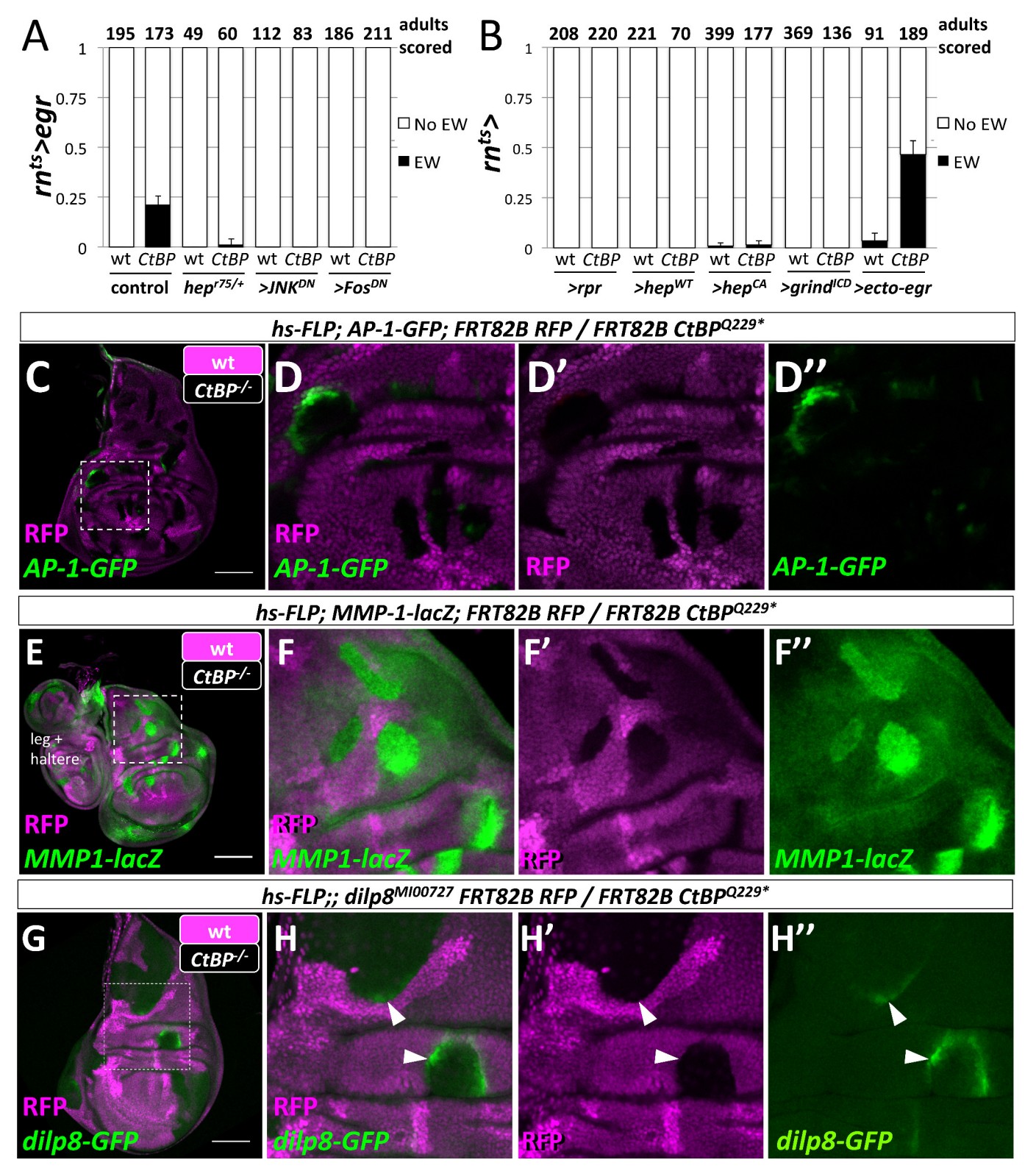

**Figure 5.** Egr activation of the JNK pathway is required for the formation of ectopic wings. (**A**) Frequency of ectopic wings (EW) following *rn^ts^>egr* damage in wild type or *CtBP^334Δ4/+^* (*CtBP^334Δ4^* was generated via CRISPR/Cas9 on the same chromosome that contains *rn^ts^>egr*) in listed genotypes: (1) *hep^r75^/+*, (2) co-expression of *UAS-JNK^DN^* and (3) co-expression of *UAS-Fos^DN^*. (**B**) Frequency of ectopic wings (EW) following *rn^ts^>* expression of listed transgenes (*UAS-rpr, UAS-hep^wt^, UAS-hep^CA^, UAS-grind^ICD^, UAS-ecto-egr*) in wild type and *CtBP^-/+^* genetic background. Temperature shifts for

*Figure 5 continued on next page*

*Figure 5 continued*

experiments shown in (A) and (B) were done on day 7 AEL for 40 hr. (C–H) Wing discs with homozygous mutant clones of *FRT82B CtBP^{Q229*}* generated by mitotic recombination with the transcriptional reporters for *AP-1-GFP* (C, D), *MMP1-lacZ* (E, F) and *dilp8-GFP* (G, H).

DOI: https://doi.org/10.7554/eLife.30391.014

The following figure supplements are available for figure 5:

**Figure supplement 1.** Co-expression of a dominant-negative Fos decreases the amount of tissue damage caused by *rn^{ts}>egr*.

DOI: https://doi.org/10.7554/eLife.30391.015

**Figure supplement 2.** Shorter ablation period does not increase the frequency of ectopic wings.

DOI: https://doi.org/10.7554/eLife.30391.016

**Figure supplement 3.** Expression of *eiger* in the wing pouch, but not in the myoblasts or notum epithelium, is sufficient to induce ectopic wings.

DOI: https://doi.org/10.7554/eLife.30391.017

**Figure supplement 4.** Manipulations of apoptosis levels during the damage period.

DOI: https://doi.org/10.7554/eLife.30391.018

**Figure supplement 5.** The regenerating wing pouch and ectopic pouch are derived from cells that express *rn-GAL4* during the ablation period.

DOI: https://doi.org/10.7554/eLife.30391.019

**Figure supplement 6.** CtBP acts outside of the *rn-GAL4* domain to prevent damage-induced ectopic wings.

DOI: https://doi.org/10.7554/eLife.30391.020

**Figure supplement 7.** *CtBP^{-/-}* mutant clones upregulate *AP-1-GFP* expression without evidence of apoptosis.

DOI: https://doi.org/10.7554/eLife.30391.021

experienced increased JNK signaling without dying quickly was needed. Such cells in the pouch could be a source of diffusible factors that promote cell fate re-specification in the notum. We observed evidence for cells that experience Egr expression and yet survive in both the regenerated wing pouch and the ectopic pouch; cells forming the ectopic pouch may activate *rn-GAL4* as they adopt a wing pouch fate (*Figure 5—figure supplement 5*).

CtBP expression was uniform in undamaged and damaged wing discs (*Figure 5—figure supplement 6*). Rescue was achieved when additional CtBP was expressed throughout the disc, but not when only expressed in the wing pouch (*Figure 5—figure supplement 6*). In addition, the knockdown of *CtBP* in the *rn-GAL4* domain during damage did not result in ectopic wings (*Figure 5—figure supplement 6*). Therefore CtBP likely functions outside the *rn-GAL4* domain, possibly in the notum, to prevent damage-induced ectopic wings.

Does a reduction in *CtBP* gene dosage affect JNK signaling? *CtBP* mutant clones in wing imaginal discs autonomously upregulated AP-1 targets *AP-1-GFP* (*Figure 5G*), *dilp8-GFP* (*Figure 5H*), and *MMP1-lacZ* (*Figure 5I*) but did not cause detectable apoptosis or elevated levels of MMP1 protein (*Figure 5—figure supplement 7*). Interestingly, the activation of *AP-1-GFP* and *dilp8-GFP* was not uniform within the mutant clones or between clones, and clones in the hinge often showed higher levels of activation. Thus, reducing CtBP protein levels could enhance JNK signaling especially in some regions of the disc.

## JAK/STAT signaling promotes the re-specification of cell fates

Since the experiments with manipulations of the JNK pathway pointed to diffusible factors that were produced in response to JNK signaling, we examined the JAK/STAT pathway. Expression of several members of the Unpaired (Upd) family of cytokine ligands are promoted by JNK signaling (*Bunker et al., 2015*). The co-expression of *UAS-upd1* with *rn^{ts}>egr* led to a dramatic increase in the frequency of ectopic wings in both wild type and *CtBP^{Q229*/+}* genetic backgrounds (*Figure 6A*), up to ~17% in wild type and ~90% in *CtBP^{Q229*/+}*. After regeneration, the wing imaginal discs that expressed *rn^{ts}>egr* and *>upd1* showed a high percent (85%) of large ectopic pouches (n = 13) (*Figure 6B*).

During normal development, JAK/STAT signaling has been reported to occur throughout the wing imaginal disc at early stages and then becomes localized to the hinge region during L2/L3 (*Ayala-Camargo et al., 2013*; *Hatini et al., 2013*; *Johnstone et al., 2013*). As shown by the reporter *STAT-DGFP*, which encodes a destabilized GFP, the activity of JAK/STAT signaling was significantly higher in early L3 discs than late L3 discs (*Figure 6C,D*). After ablation of the pouch using *rn^{ts}>egr* in wild-type discs, *STAT-DGFP* expression was observed immediately surrounding the ablated pouch (*Figure 6E*), as observed previously, (*La Fortezza et al., 2016*), with very little STAT-

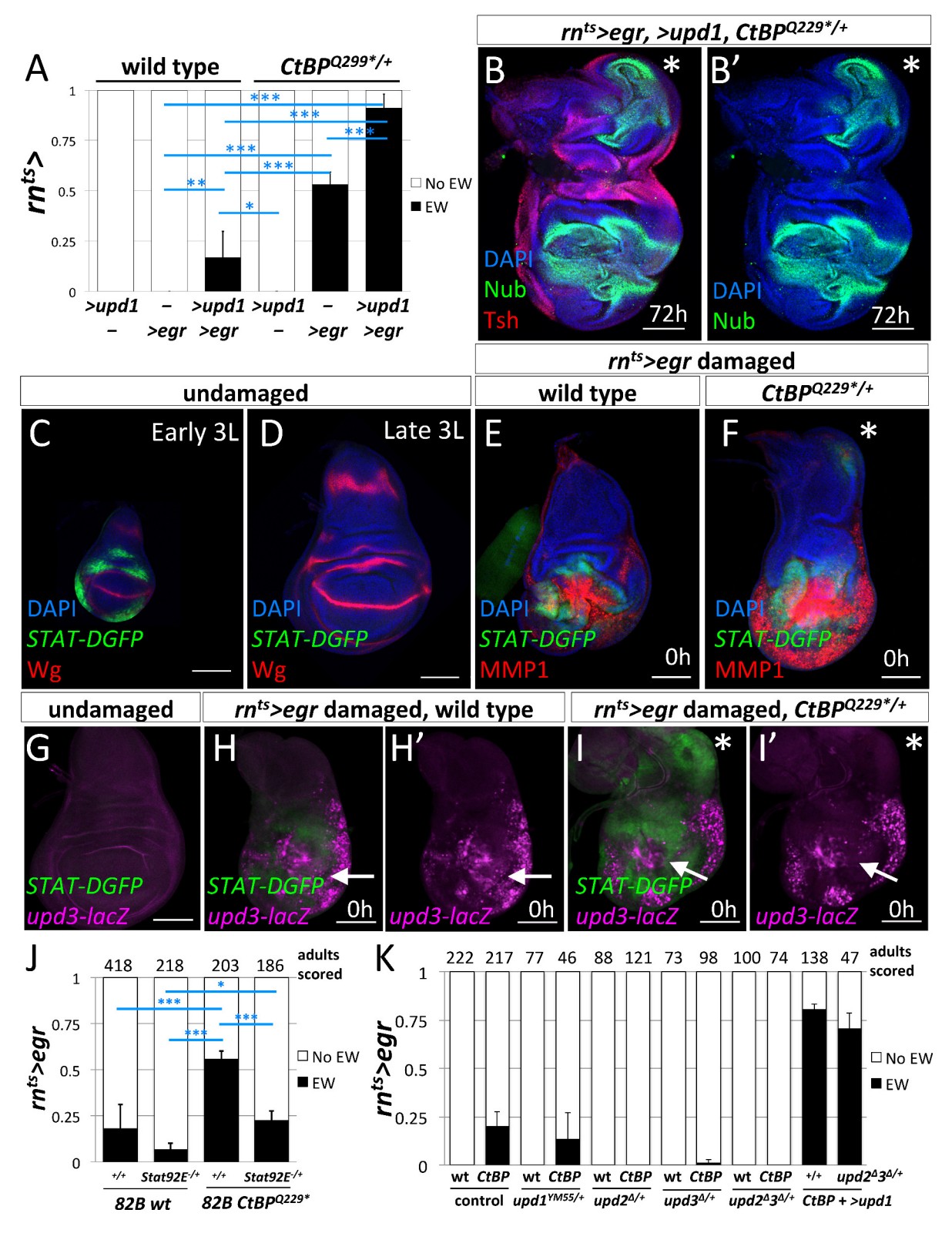

**Figure 6.** JAK/STAT signaling is important for the formation of the ectopic wings. (**A**) Frequency of ectopic wings (EW) following $rn^{ts}>$ driven expression of $>upd1$, $>egr$ or $>upd1$ together with $>egr$ in wild type and $CtBP^{Q229*/+}$discs. Data were compared using ANOVA followed by Tukey test for significance (*p<0.05, **p<0.01, ***p<0.001) (**B**) Wing disc following $rn^{ts}>$ driven co-expression of $>egr$ and $>upd1$ after 72 hr of recovery. Pouch identity is shown by anti-Nub and an asterisk marks the ectopic pouch. (**C–D**) Early L3 (**C**) and late L3 (**D**) wing discs stained with anti-Wg and anti-GFP

*Figure 6 continued on next page*

*Figure 6 continued*

to detect the fast turnover *STAT-DGFP* reporter. (E–F) Wing discs with *STAT-DGFP* reporter following *rn^{ts}>egr* damage at 0 hr of recovery in wild type (E) and *CtBP^{Q229*/+}* (F) stained with anti-MMP1. Note expression of MMP1 in a subset of cells expressing *STAT-DGFP* in the area of ectopic pouch formation (asterisk); see (*Figure 6—figure supplement 1*). (G–I) Wing imaginal discs with both *upd3-lacZ* and *STAT-DGFP* reporters in undamaged (G) and damaged wild type (H) and *CtBP^{Q229*/+}* (I) discs at 0 hr of recovery. Note area in the notum that expresses *STAT-DGFP* but does not have elevated levels of *upd3-lacZ*. (J) Frequency of EW following *rn^{ts}>egr* damage when crossed to 1) *FRT82B*, 2) *FRT82B Stat92E^{85C9}*, 3) *FRT82B CtBP^{Q229*}*, and 4) *FRT82B CtBP^{Q229*} Stat92E^{85C9}*. Data were compared using ANOVA followed by Tukey test for significance (*p<0.05, **p<0.01, ***p<0.001). (K) Frequency of EW following *rn^{ts}>egr* damage in wild type and *CtBP^{334Δ4/+}* in the listed genotypes (1) +/+, (2) *upd1^{YM55}/+*, (3) *upd2^Δ/+*, (4) *upd3^Δ/+*, and (5) *upd2^Δupd3^Δ/+*. In addition, co-expression of *>upd1* with *rn^{ts}>egr* in (1) *CtBP^{334Δ4/+}* and (2) *upd2^Δupd3^Δ/+;; CtBP^{334Δ4/+}* genetic backgrounds. Note the rescue of EW induction in *upd2^Δupd3^Δ/+* background by the co-expression of *>upd1*.
DOI: https://doi.org/10.7554/eLife.30391.022

The following figure supplements are available for figure 6:

**Figure supplement 1.** *STAT-DGFP* expression in the notum of damaged wing discs is independent of MMP1 expression.
DOI: https://doi.org/10.7554/eLife.30391.023

**Figure supplement 2.** Upd1 overexpressed in the myoblasts can act on the disc proper epithelium and disrupt the notum Wg stripe.
DOI: https://doi.org/10.7554/eLife.30391.024

**Figure supplement 3.** *CtBP^{-/-}* mutant clones do not alter expression of *upd3-lacZ*.
DOI: https://doi.org/10.7554/eLife.30391.025

**Figure supplement 4.** Effect of reducing CtBP function on STAT activity.
DOI: https://doi.org/10.7554/eLife.30391.026

*DGFP* expression in the notum. In contrast, in many *CtBP^{Q229*/+}* discs, a patch of *STAT-DGFP* expression was detected in the notum (*Figure 6F*), suggesting that JAK/STAT signaling may play a role in inducing a cell fate change. *STAT-DGFP* expression was often observed around sites of MMP1 expression. However, we observed that in some *CtBP^{Q229*/+}* discs, only *STAT-DGFP* expression was detected, without a spot of MMP1 expression, suggesting that JAK/STAT activity in the notum can occur independently of ectopic JNK activation (*Figure 6—figure supplement 1*).

If JAK/STAT activation occurs in the notum following damage to the pouch, where is the ligand being produced? It is known that Unpaired-family ligands are upregulated following damage to discs (*Pastor-Pareja et al., 2008*) and during regeneration (*Katsuyama et al., 2015*; *Santabárbara-Ruiz et al., 2015*; *La Fortezza et al., 2016*). We used the reporter *upd3-lacZ* (*Bunker et al., 2015*), which is not expressed at high levels in undamaged late L3 discs, to investigate where Upd3 ligand is being produced (*Figure 6G*). At 0 hr of recovery, *upd3-lacZ* expression was observed in the region of the ablated pouch in both wild type (*Figure 6H*) and *CtBP^{Q229*/+}* discs (*Figure 6I*), but not in the notum, even though *STAT-DGFP* expression was detected there (*Figure 6I'*). Since ligands of the Upd family are known to diffuse considerable distances, it is possible that the Upd ligands were generated by tissue damage and JNK activation in the pouch region and then diffused to the notum and activated JAK/STAT signaling. Indeed, the expression of *UAS-upd1* in the myoblasts, which are basal to the notum epithelium, caused the activation of the JAK/STAT pathway in the notum and the disruption of the notum Wg stripe, but not the formation of an ectopic pouch (*Figure 6—figure supplement 2*). A similar disruption of the notum Wg stripe was observed following *rn^{ts}>egr* damage (*Figure 3D*). Thus disruption of the notum Wg stripe appears insufficient to trigger ectopic pouch formation and other events seem necessary.

If JAK/STAT activation is needed to form ectopic wings, then a decrease in JAK/STAT activity may suppress the frequency of ectopic wings. The removal of one copy of the JAK/STAT transcription factor *Stat92E* (*FRT82B Stat92E^{85C9}*) significantly decreased the frequency of ectopic wings (EWs) when compared to *FRT82B* (which had a background rate of EWs) in both a wild type and *CtBP^{Q229*/+}* genetic background (*Figure 6J*). The reduction in the levels of the Upd-family ligands, by removal of one copy of either *upd2* or *upd3*, or both *upd2* and *upd3* significantly reduced the frequency of ectopic wings (*Figure 6K*). In addition, this suppression was completely rescued by the addition of *UAS-upd1*, which indicates that the different Upd-family ligands are acting similarly in this situation. Together, this strongly indicates that the levels of JAK/STAT signaling can modulate the frequency of the notum-to-pouch cell fate change.

Does *CtBP* modulate the JAK/STAT pathway? First, we tested if *upd3* expression was altered in *CtBP^{-/-}* clones and found no change under undamaged conditions (*Figure 6—figure supplement 3*).

Then we tested the *STAT-DGFP* reporter and found a slight increase in STAT signaling, although only in young discs and in regions where the pathway is normally active (*Figure 6—figure supplement 4*). Thus reducing CtBP levels may make tissues more responsive to the JAK/STAT pathway.

## Why does the cell-fate respecification occur at a specific location in the notum?

The term 'weak point' has been used to describe regions prone to fate change or transdetermination, (reviewed in [*McClure and Schubiger, 2007*]). This might be because specific parts of the disc are more sensitive than others to the effects of Upd family ligands. Consistent with this notion, the expression of *UAS-upd1* along the A-P compartment boundary specifically activated *dilp8-GFP* expression in the notum, in a similar location to the observed fate change in the notum (*Figure 7B*). In addition, this same area in the notum was specifically responsive to *UAS-upd1* produced by random GAL4-expressing clones throughout the disc (*Figure 7C*), as shown by strong *dilp8-GFP* expression in this notum region (*Figure 7D–F*). *dpp>upd1* only activated *STAT-DGFP* in specific regions of the notum and wing pouch, although there was an increase in apoptosis throughout the wing disc and adults showed notum defects consistent with a loss of proper notum patterning (*Figure 7—figure supplement 1*). Taken together, these experiments indicate that there is a region in the notum of the wing disc that is specifically and highly responsive to Upd ligands. However, the addition of *UAS-upd1* without activation of JNK did not seem to be sufficient to trigger a fate change suggesting that JNK signaling and JAK/STAT signaling function together to promote respecification of cell fates.

## Discussion

The replacement of damaged tissue by the proliferation of adjacent cells requires those cells to change fates, often in subtle ways. In the imaginal disc, cells at each location have distinct patterns of gene expression. Hence, if portions of the disc are eliminated and replaced by the proliferation of nearby cells, the newly generated cells will need to adopt characteristics distinct from the cells that they are derived from. At the same time, other cells need to preserve established patterns of gene expression. Our work suggests that the limited plasticity that is necessary for regeneration results from an interplay of pathways that promote plasticity such as JNK/AP-1 and JAK/STAT with regulators that limit the activity of these pathways such as CtBP.

### The relationship between ablation of the pouch and activation of the JNK/AP-1 and JAK/STAT in the notum

Activation of two pathways, JNK/AP-1 and JAK/STAT, appears to be necessary for promoting cells in the notum to adopt a pouch fate. How does ablation of the pouch promote the activation of these pathways in the notum? Neither ablation of the pouch using *rpr* nor activation of the Egr-receptor Grind in the pouch suffices. Thus, Egr is necessary and it either activates additional Grnd-independent pathways, or, more likely, activation of the Egr pathway in cells outside the *rn-GAL4* expression domain is necessary. This could be both to activate JNK/AP-1 signaling locally around the damaged pouch and at a greater distance in the notum. The local signal to the hinge cells surrounding the wing pouch may produce the *upd* ligands, as these cells normally express *upd1*. Egr may also signal directly to the notum, as a cleaved secreted ligand or from membrane fragments of cellular debris, much of which is trapped apically. In support of this idea, a secreted form of Egr is more effective than the membrane tethered form at inducing ectopic wings. Since reducing *CtBP* function increases AP-1 activity, the notum of $CtBP^{-/+}$ discs will likely be more responsive to Egr signaling, leading to the activation of downstream AP-1 targets, such as *dilp8* and *wg*.

$\quad$*STAT-DGFP* expression in the notum is increased in *CtBP* heterozygotes following ablation of the pouch. Increased JNK activity in the dying cells of the pouch would be expected to generate increasing levels of Upd-family ligands since they are upregulated in regenerating discs (*Katsuyama et al., 2015*; *Santabárbara-Ruiz et al., 2015*; *La Fortezza et al., 2016*) and regulatory elements of these genes have AP-1-binding sites (*Bunker et al., 2015*). Consistent with their production by dying cells is the observation that more rapid killing of pouch cells by co-expression of *rpr* reduces notum-to-pouch transformations.

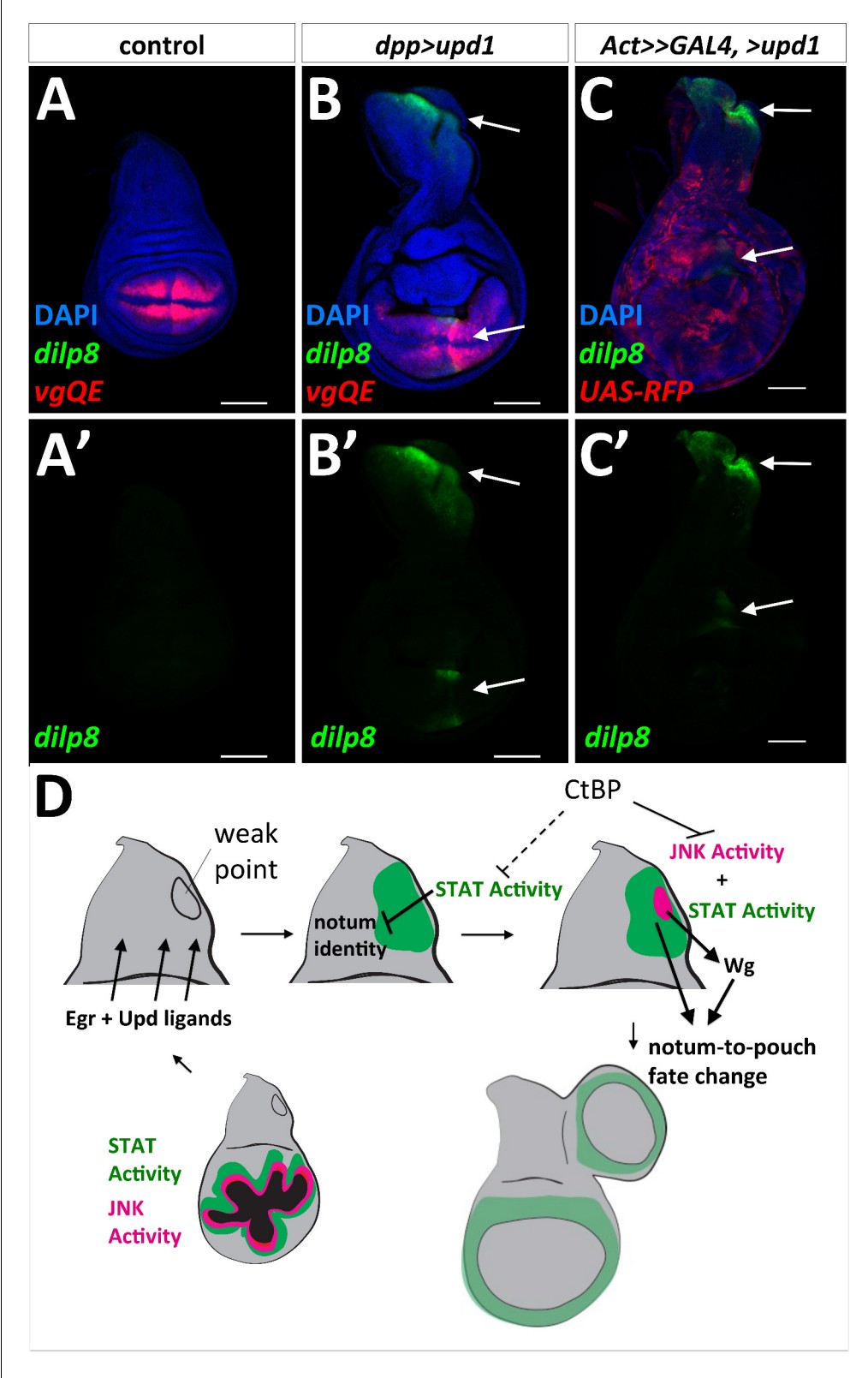

**Figure 7.** 'Weak point' in the wing imaginal disc is responsive to Upd1 ligand and model for notum to wing transdetermination. (A, B) Wing discs with *dilp8-GFP* and *vgQE-RFP* reporters: control (A), and *dpp-GAL4, UAS-upd1* (B). Note the specific expression of *dilp8-GFP* along the posterior edge of the notum and in the wing pouch (arrows). (C) Overexpression of *UAS-upd1* by heat-shock induced flip-out clones (*hs-FLP, Act<stop<GAL4*) marked with *UAS-RFP*. Note the specific area in the notum that responds to Upd1 by expressing *dilp8-GFP* (arrows). (D) Model for how JAK/STAT and JNK/AP-

*Figure 7 continued*

1 signaling work together to trigger a fate change and the growth of an ectopic pouch. Damage to the normal wing pouch generates both Egr and Upd ligands. The weak point, an area near the anterior/posterior compartment boundary in the notum responds to Upd ligands and the resulting JAK/STAT signaling disrupts the normal notum identity (as observed with the disruption of the notum Wg stripe). JNK activity within a field of JAK/STAT activity leads to the expression of Wg, which triggers cells to adopt the new wing pouch fate. Then, as in the regenerating wing pouch, additional cells are recruited to generate a new pouch. CtBP functions, either directly or indirectly, to antagonize these pathways.

DOI: https://doi.org/10.7554/eLife.30391.027

The following figure supplement is available for figure 7:

**Figure supplement 1.** The notum and wing pouch respond to the overexpression of *UAS-upd1* by *dpp-GAL4*.

DOI: https://doi.org/10.7554/eLife.30391.028

Increased STAT activity in the notum reduces notum-specific patterns of gene expression and promotes a hinge-like fate (*Ayala-Camargo et al., 2013*). Consistent with this, we found that reducing gene dosage of *mirr*, a homeodomain-containing protein expressed in the notum, increased the frequency of ectopic wings. A recent study reported the generation of ectopic wings simply by expression of *UAS-upd1* in the notum (without directed tissue damage) (*Recasens-Alvarez et al., 2017*). However, this study induced expression in the second larval instar (day 5 at 18°C) and did not report if expression of *UAS-upd1* at later time points could also induce ectopic wings. Of note, we have found that expression of high levels of *UAS-upd1* generates considerable levels of apoptosis. Conceivably, the generation of ectopic wings in that report could therefore also have resulted from the combined activation of the JAK/STAT and AP-1 pathways.

## Why does formation of the ectopic pouch begin at a specific location in the notum?

Imaginal discs have been hypothesized to have 'weak points', where transdetermination occurs most often following physical fragmentation (*Schubiger, 1971*), (reviewed in [*McClure and Schubiger, 2007*]). Other cell fate transformations are also more likely to occur in particular regions of the discs (*Salzer and Kumar, 2010*). For leg-to-wing transdetermination in the prothoracic leg disc, the weak point is located where two different morphogens, Wg and Dpp, are co-expressed (*Johnston and Schubiger, 1996*; *Maves and Schubiger, 1998*). In the wing disc, Dpp is expressed along the anteroposterior (AP) compartment boundary and Wg, is expressed along part of the dorsoventral (DV) compartment boundary through the wing pouch. The AP and DV boundaries likely intersect at two locations – one in the middle of the wing pouch and in the notum near the region where we observe ectopic pouch formation. In addition, the notum Wg stripe in the L3 disc generates an additional location where both Wg and Dpp are expressed. Therefore, it is possible that similar to the weak point in the leg disc, the presence of both these morphogens makes cells more prone to change fate, especially following damage.

This region of the notum could be predisposed to an increase in STAT activity. When *UAS-upd1* is expressed using *dpp-GAL4*, *STAT-DGFP* expression is not observed along the entire GAL4 domain. Rather, it is mostly restricted to the pouch, and importantly, a region of the notum. These regions are more responsive to Upd-family ligands than other parts of the disc and the area in the notum is likely the weak point for notum-to-pouch fate change. Recent studies have described regions in wing discs that are tumor hot-spots (*Tamori et al., 2016*) and are resistant to apoptosis (*Verghese and Su, 2016*) and found that they also have elevated JAK/STAT activity. Thus, parts of the disc that have elevated STAT activity may be more capable of surviving damage and then re-specifying cell fates.

## Events during the formation of an ectopic pouch have similarities to specification of the pouch during normal development

During normal development, the initial specification of the wing pouch occurs towards the end of the first larval instar. At this time STAT is active through most of the wing disc and *wg* is expressed in a patch in the anterior and ventral portion of the disc. Subsequently, *wg* expression expands from its initial region of expression when cells surrounding the nascent pouch are recruited to a pouch fate by a feed-forward mechanism that is fueled by Wg (*Wu and Cohen, 2002*; *Zecca and Struhl, 2007b*, *2010*). In the absence of Wg expression, the pouch is not specified.

In our system, *wg* expression from the damage-responsive enhancer is important for triggering ectopic wings. However, we found that additional Wg (*rn^{ts}*>egr, >wg), if at all, decreased the frequency of ectopic wings and that Wg alone was incapable of transforming a more mature notum to an ectopic pouch; therefore JAK/STAT and AP-1 activity appear necessary. Based on our observations and those of others, JAK/STAT activity can reduce the expression of notum-specific genes (*Ayala-Camargo et al., 2013*). This likely transforms that part of the notum to a less committed state. Thus, a region with elevated STAT activity leads to relatively uncommitted notum cells and within this region, the activation of AP-1 promotes Wg expression via a damage-responsive enhancer, which may create conditions similar to the initial specification of the wing pouch at earlier stages of development (*Figure 7D*).

Once the ectopic pouch has been specified, the nascent pouch could turn on the *rn-GAL4* driver, even briefly, prior to the downshift. Then *egr* expression would occur resulting in localized apoptosis and JNK activation. The surrounding cells would activate AP-1 and STAT and also express *wg* as occurs in the regeneration blastema surrounding *rn^{ts}*>egr ablated tissue. This would then have the effect of reinforcing the conditions that are necessary for notum-to-pouch fate change and the generation of a new wing pouch. While this paper was under review, another paper was published that gives support to damage-induced fate changes relying on JAK/STAT and Wg signaling (*Verghese and Su, 2017*).

### Role of CtBP in resisting cell fate changes

We have shown that clones of cells lacking CtBP function have elevated levels of AP-1 activity and, in a few cases in specific portions of the disc, increased STAT activity. CtBP is a co-repressor that interacts with a number of transcriptional repressors via a PxDLS motif including Brinker (*Hasson et al., 2001*) and Hairy (*Poortinga et al., 1998*) and hence multiple CtBP-containing complexes could be relevant for this role of CtBP. Intriguingly, some of the isoforms of the *Drosophila* Fos ortholog, Kayak, which heterodimerizes with Jun to form the AP-1 transcription factor, have PxDLS motifs. While AP-1 has mostly been studied as a transcription factor that activates gene expression upon phosphorylation of Jun by JNK, in the absence of sufficient activating signals, it may repress inappropriate expression of AP-1 target genes via CtBP. In this way, CtBP could buffer cell fates from destabilization caused by fluctuations in the level of JNK activity.

## Materials and methods

### Fly stocks

The stocks that were used in this study include: *rn-GAL4, tub-GAL80^{ts}, rn-GAL4, tub-GAL80^{ts}, UAS-egr* and *rn-GAL4, tub-GAL80^{ts}, UAS-rpr* (*Smith-Bolton et al., 2009*), *hh-GAL4, ptc-GAL4, tub-GAL80^{ts}, nub-GAL4; tub-GAL80^{ts}, UAS-egr/TM6B-GAL80, dpp-GAL4* (BL:1553), *R15B03-GAL4* (BL:49261), *R76A01-GAL4* (BL:46953) (*Pfeiffer et al., 2008*), *G-TRACE* (BL:28280, BL:28281) and *Ubi<stop<GFP^{nls}* (BL:32250, BL:32251)(*Evans et al., 2009*), *Act5C>FRT.CD2>GAL4, UAS-RFP* (BL:30558), *vgQE-RFP* (*Zecca and Struhl, 2007a*), *hs-FLP; Act<stop<lacZ^{nls}, Ubi<stop<GFP^{nls}* (*Worley et al., 2013*), *UAS-CtBP* (*Bhambhani et al., 2011*), *Genomic CtBP construct (gCtBP)* (*Zhang and Arnosti, 2011*), *UAS-ecto-eiger* (*Narasimamurthy et al., 2009*), *UAS-grind^{ICD}* (*Andersen et al., 2015*), *MMP1-LacZ* (*Uhlirova and Bohmann, 2006*), *AP-1-GFP* and *AP-1-RFP* reporters (*Chatterjee and Bohmann, 2012*), *upd3-lacZ* (*Bunker et al., 2015*), *upd3.1-lacZ* (*Jiang et al., 2011*), *BRV-B-GFP* (*Harris et al., 2016*), *UAS-Dronc^{DN}* (*UAS-Dronc^{C318S}*) (*Hawkins et al., 2000*), *FRT82B, FRT82B CtBP^{Q229*}* and *FRT82B Stat92E^{85C9}*. Stocks obtained from Bloomington stock center include: *CtBP^{87De-10}* (BL:1663), *CtBP^{03463}* (BL:11590), *UAS-CtBP^{RNAi}* (BL:31334, BL:32889), *mirr^{1486}* (BL:23935), *mirr^{1825}* (BL:23928), *hep^{r75}* (BL:6761), *UAS-Fos^{DN}* (*UAS-Fra.Fbz*, BL:7214), *UAS-JNK^{DN}* (*UAS-bsk^{K53R}*, BL:9311), *UAS-p35* (BL:5072, 5073), *dIlp8^{MI00727}* (BL:33079), *UAS-wg* (BL:5918, BL:5919), *UAS-hep^{CA}* (BL:6406), *lexAOp-FLP* (BL:55820), *R81E08-lexA* (BL:54377), *R76B02-lexA* (BL:54118), *R76B06-lexA* (BL:54225), *R15C03-lexA* (BL:52490), *10XSTAT92E-GFP* (BL:26197) and *10XSTAT92E-DGFP* (BL:26199) (*Bach et al., 2007*), *Df(3R) Exel8157* (BL:7973), and *upd2^{Δ}* (BL:55727), *upd2^{Δ}upd3^{Δ}* (BL:55729), *upd3^{Δ}* (BL:55728)(*Osman et al., 2012*).

## Temperature shift and clone induction experiments

Genetic ablation experiments were conducted as described in (*Smith-Bolton et al., 2009*). Briefly, eggs were collected on grape plates and animals were density controlled by picking 55 L1 larvae into vials with yeast paste. Shifts were conducted on particular days (day 3 - day 9) AEL from an 18°C incubator to 30°C incubator. Cultures were shifted on day 7 AEL for 40 hr unless otherwise noted. Experimental replicates were done on separate days. Adults were scored as (1) No EW for no ectopic wings or (2) EW for either one or two ectopic wings, except in (*Figure 3*), where adults were scored for having zero, one, or two ectopic wing(s). Graphs show average frequency of EWs for experiments done on separate days. Error bars show standard deviations between the experiments.

To generate *CtBP* mutant clones, *hs-FLP;;FRT82B RFP$^{nls}$* was crossed to *FRT82B CtBP$^{Q229*}$* and vials were heat shocked at 37°C for 1 hr to induce mitotic clones. To generate randomly marked clones, *hs-FLP;;rn$^{ts}$>egr/TM6B, Tub-GAL80* was crossed to *Act5C<stop<lacZ$^{nls}$, Ubi<stop<GFP$^{nls}$; CtBP$^{Q229*}$/TM6B* and the cultures were staged as described above. A brief 10 min 37°C heat-shock was used to generate flip-out clones at a relatively low density. Without a 37°C heat shock, the 30°C temperature shift for the genetic ablation experiments did not produce *hs-FLP* induced clones, which we also noted in (*Smith-Bolton et al., 2009*).

## Immunohistochemistry

Imaginal discs were fixed in 4% paraformaldehyde for 15 min, washed and permeabilized in phosphate-buffered saline solution with 0.1% Triton X-100, and blocked in either 10% Normal Goat Serum or Normal Donkey Serum depending on staining. The following antibodies were used from Developmental Studies Hybridoma Bank (DSHB): mouse anti-Wg (1:100, 4D4); mouse anti-Mmp1 (1:100, a combination of 14A3D2, 3A6B4 and 5H7B11); mouse anti-En (1:10, 4D9); rat anti-Ci (1:10, 2A1); mouse anti-Cut (1:200, 2B10); mouse anti-Ptc (1:50, Apa-1); and mouse anti-Ubx (1:20, FP3.38). The following antibodies were gifted: rat anti-Pdm2 (Chris Doe); rat anti-Twist (1:1000, Eric Wieschaus); and mouse anti-Nub and rabbit anti-Tsh (Stephen Cohen). The following antibodies are from commercial sources: goat anti-dCtBP (1:50, dN-20 Santa Cruz Biotechnology, Dallas TX); rabbit anti-DCP-1 (1:250, Cell signaling); rabbit anti-GFP (1:500, Torrey Pines Laboratories, Secaucus, NJ); chicken anti-GFP (1:500, ab13970 Abcam, Cambridge, UK); and rabbit anti-β-galactosidase (1:1000, #559762; MP Biomedicals, Santa Ana, CA). Secondary antibodies were from Cell Signaling. Nuclear staining was by DAPI (1:1000, Cell Signaling). Images were obtained on either a Leica TCS or a Zeiss LSM 700. Images were processed using ImageJ (Fiji) (*Schindelin et al., 2012*). All scale bars are 100 μm.

For scoring discs with *dilp8-GFP* expression in the notum or ectopic pouch at different time points of recovery, in (*Figure 3*), images were collected from discs from biological replicates (cultures started on separate days) and imaged on Zeiss Axio Imager M1 without staining with anti-GFP. Images were collected and then scored for the presence or absence of a second spot of *dilp8-GFP* expression.

## Generation of *CtBP* CRISPR alleles

We generated CRISPR guide RNAs to regions conserved in all *CtBP* isoforms, using the method described in FlyCas9 system (*Kondo and Ueda, 2013*). We generated stable transgenetics using the U6-promoter driving the guide RNAs inserted at the *attP40* landing site.The guide RNAs were optimized to work in our *wt$^{iso}$* genetic background, as we sequenced the predicted regions guide RNA binding, detected single nucleotide polymorphisms as compared to the FlyBase consensus, and incorporated these changes into the guide RNA sequence. We used the transgenic services provided by BestGene (Chino Hills, CA). We generated the following stocks to generate CRISPR mutations on particular third chromosomes: *Sp/Cyo, nos-Cas9; wt$^{iso}$* and *Sp/Cyo, nos-Cas9; rn-GAL4, UAS-eiger, tub-GAL80$^{ts}$*, which were crossed to *U6-CtBP-guideRNA; TM2/TM6B* in order to recover CRISPR generated mutations on particular third chromosomes. Single males carrying possible *CtBP* alleles were tested for the failure to complement viability when crossed to a deletion that spans the *CtBP* gene, *Df(3R)Exel8157*. The newly generated *CtBP* alleles were then sequenced to determine the nucleotide change near the guide RNA cut site.

## Acknowledgements

The authors would like to thank D Arnosti, E Bach, K Basler, K Cadigan, C Doe, S Cohen, P Leopold, G Struhl, E Wieschaus, and M Zeidler for stocks and reagents. We thank current and former members of the Hariharan lab for useful feedback, especially T Sumabat and L Setiawan. We thank D Bilder and the Bilder lab members for advice. We thank E Bondra, K Gandhi, and V Dhruva for technical assistance. We thank the Bloomington Stock Center, Drosophila Genomics Resource Center, Developmental Studies Hybridoma Bank, and BestGene for stocks, reagents, and services. This work was funded by NIH grants R01 GM061672, R01 GM085576, and R35 GM122490 and a Research Professor Award from the American Cancer Society (RP-16238–06-COUN).

## Additional information

### Funding

| Funder | Grant reference number | Author |
| --- | --- | --- |
| National Institute of General Medical Sciences | GM061672 | Iswar K Hariharan |
| American Cancer Society | RP-16-238-06-COUN | Iswar K Hariharan |
| National Institute of General Medical Sciences | GM085576 | Iswar K Hariharan |
| National Institute of General Medical Sciences | GM122490 | Iswar K Hariharan |

The funders had no role in study design, data collection and interpretation, or the decision to submit the work for publication.

### Author contributions

Melanie I Worley, Conceptualization, Data curation, Formal analysis, Supervision, Validation, Investigation, Visualization, Methodology, Writing—original draft, Writing—review and editing; Larissa A Alexander, Data curation, Investigation; Iswar K Hariharan, Conceptualization, Supervision, Funding acquisition, Writing—original draft, Project administration, Writing—review and editing

### Author ORCIDs

Melanie I Worley ◉ http://orcid.org/0000-0001-9772-4985
Iswar K Hariharan ◉ http://orcid.org/0000-0001-6505-0744

### Decision letter and Author response

Decision letter https://doi.org/10.7554/eLife.30391.032
Author response https://doi.org/10.7554/eLife.30391.033

## Additional files

### Supplementary files

• Supplementary file 1. Screen summary for third chromosome deletions. The frequency of ectopic wings (EW) following $rn^{ts}$>egr damage for third-chromosome deletions screened. The deletions are ordered based on the cytogenetic position along the third chromosome.
DOI: https://doi.org/10.7554/eLife.30391.029

• Transparent reporting form
DOI: https://doi.org/10.7554/eLife.30391.030

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
