## [Decision Letter]

Thank you for submitting your article "The transcriptional co-repressor CtBP impedes cell fate misspecifications in regenerating *Drosophila* imaginal discs" for consideration by *eLife*. Your article has been reviewed by three peer reviewers, one of whom, Bruce Edgar (Reviewer #1), is a member of our Board of Reviewing Editors, and the evaluation has been overseen by Didier Stainier as the Senior Editor. The following individuals involved in review of your submission have agreed to reveal their identity: Giovanni Bosco (Reviewer #3).

The reviewers have discussed the reviews with one another and the Reviewing Editor has drafted this decision to help you prepare a revised submission.

Reviewing Editor's comments:

All three reviewers appreciated the approaches used in this paper, and its content. However reviewer #1 felt that the paper was much too long and had many inconclusive sections that could be condensed or removed. In discussions, the other reviewers agreed with this perspective.

Reviewer #3 also felt that the paper should have delivered more mechanistic insight as to how CtBP affects transdetermination, but we understand that providing this could be challenging and time-consuming, and so we decided that it is not required at this stage. Hence we would like to invite you to submit a revised manuscript that is significantly shorter than the current version (30-50%) and more linear in presentation. Specific suggestions for how to do this can be found in the reviews, which I include below essentially in their entirety.

In addition, the points #4 and #6 from reviewer 1 merit special consideration – if you can provide further data addressing these points this would be much appreciated. At the very least, a clear discussion of the action of *egr* is needed. The reviewers also had a number of minor comments regarding controls, clarification of methods and interpretation of the results, which you should address in your revision. Overall we agreed that the paper was sufficiently developed and could be published without large amounts of additional data, provided the delivery is succinct and clear. *eLife* looks forward to receiving your revision.

Reviewer #1:

In this paper, Worley, Alexander, and Hariharan revisit the phenomenon of damage-initiated trans-determination in *Drosophila* wing imaginal discs, using new genetic methods to identify the genes involved. This is an interesting process with potentially broad relevance, but little progress has been made on it in years. The authors present a very nice genetic saturation screen of the third chromosome (~40% of the genome) that identifies many genes potentially involved. Then they focus on one gene, CtBP at 87D, and show how this affects transdetermination of notum cells to wing pouch cells. The process involves JNK and STAT activation (as expected) at the wound site, is multi clonal, and involves the activation of a number of other genes that are tracked. The effects of these genes are are clearly demonstrated, but the paper falls short of determining any clear mechanisms at the molecular level.

This manuscript starts out nicely with one of the best introductions on the subject of transdetermination that I've read, and succinct descriptions of the damage protocol (already detailed in a previous publication) and the modifier screen, which is clever and effective (Figure 1). The effect of CtBP is clear, but it was not obvious why the authors focused on this one gene, and this should be stated. Following this nice entry, the paper details a huge number of genetic tests of mechanism in the subsequent 7 figures and 15 pages of meandering text. The specific tests that were done are all logical, and the results are interpreted in a critical fashion, but unfortunately they don't come together in a very readable, coherent form. Although the effects of *egr*, Ap1, Stat, etc are clear, the authors don't succeed in demonstrating how these factors work together very well, and some obvious questions are left unresolved (see below). In addition, many tests that were relatively inconclusive are presented, as well as quite a few negative results and dead ends. Many paragraphs in the Results section don't have a conclusion at all. Overall the Results section meanders from one issue to the next, with too much discussion, and no obvious direction. It is far too long. It seems to be a very raw draft of a large body of work, essentially un-refined, and it really should be re-written in a much more concise, linear format to be digestible by a general audience. Much of what is here could probably be presented much more simply, or just omitted, without affecting the paper's final conclusions. I suggest trying to pare it down to about half the current length. While I think the core content of this paper is valuable and should be published, I cannot recommend it in its current form. Specific points follow below:

1) There is a long section about the Rga locus but then this locus is never mentioned again. What is it's map location? What is its relevance to the paper's overall conclusion? This locus justifies the screen, but if it is not relevant to the paper's conclusions the section devoted to it should be removed. The paper contains several other examples of discussion of this sort of irrelevant, distracting data.

2) The authors should note why they focused on region 87D after doing the screen. Further discussion of other loci that strongly affected transdetermination could enhance this paper, if the genes involved are known. Nevertheless, it is very nice that the authors include the complete results of their very elegant screen (Figure 1, Supplementary file 1).

3) The long paragraph in subsection “Re-specifying cell fates requires the activation of the JNK/AP-1 pathway” is an example of a discussion of results that meanders and has no conclusion.

4) There are quite a few experiments that address the requirements of *egr*, Jnk, and apoptosis in promoting transdetermination, and much discussion about this. However the paper doesn't seem to have a clear conclusion about whether *egr* triggers transdetermination via JNK-mediated AP1 activation, or JNK-mediated apoptosis, or both. Egr is clearly sufficient, and *JNK* required but not sufficient, and apoptosis is not sufficient, but it's not clear what, downstream of *egr*, is doing it. Is apoptosis required? I think a few more experiments are required here, and a cleaner more definitive conclusion is needed about which downstream targets of *egr* are critical.

5) The effects on dILP8 expression are interesting and suggestive (“dilp8 expression in a secondary location is predictive of ectopic wing pouch formation”), but dILP8 function is not tested. It would be a nice addition to the paper if the role of dILP8 were tested using mutants, RNAi, and overexpression.

6) Upd ligand production, in response to *egr*, or Ap1, or apoptosis directly, seems to be an important aspect of the authors proposed mechanism (Figure 8Q, and Results). However they don't seem to have tested the requirement functionally, except by *Upd1* overexpression and in Stat+/- mutants. There are *upd2, upd3*, and *upd2,3* deletion lines that are viable and could be used to test the requirement of these damage-dependent ligands.

7) Given the multitude of tests presented, the summary diagram (Figure 8Q) is surprisingly simple. Could the authors add more detail to this? Alternatively, the paper could be much more readable if it included ONLY results that are directly relevant to this model.

Reviewer #2:

The manuscript by Worley et al. shows the requirement of the transcriptional co-repressor CtBP for proper cell fate specifications in regenerating *Drosophila* larval wing imaginal discs after apoptosis-induced damage. The authors first find that, in certain genetic backgrounds, regeneration after *rn^ts^*>*egr* damage produces ectopic wing tissues. They then conduct a genetic screen using stocks bearing deletions of the third chromosome in the *rn^ts^*>*egr* background and find that many deletions induce the formation of ectopic wings at varied penetrance. They then identify and focus on CtBP, a transcriptional co-repressor. Regarding the signaling pathway, the authors demonstrate that Egr-dependent JNK activation in the wing pouch is necessary for the generation of ectopic wings and that JAK-STAT signaling promotes the notum-to-pouch transformation. They further find that cells in the notum are re-specified to ectopic wing pouch, but maintain their compartment identity, show that epigenetic regulation plays an opposite role to STAT and Wg in modulating the frequency of ectopic growth. They finally show that loss of CtBP increases the AP-1 activity and propose a model to explain how CtBP prevents cell fate specifications in regenerating *Drosophila* imaginal discs.

The results presented in this manuscript are very interesting and will be of interest to the field of tissue regeneration and developmental signaling. The authors conduct a series of elegant genetic assays and provide compelling results to validate the conclusions they made, therefore the conclusions are convincing.

Specific Comments:

1) Figure 4. Since temperature shift and genetic background affect the frequency of ectopic wings, in order to conclude that "expression of >*egr* in the wing pouch is the major cause of generating ectopic wings", the authors also need to test the R15B03-, R76A01-, and dpp-Gal4 drivers using the same temperature shift protocol.

2) Scale bar is missing in some figures.

3) Since the clean genetic background did not produce ectopic wings in *rn^ts^*>*egr*, the description at the beginning part of the manuscript is misleading. This information should be included in the Abstract.

4) *rn^ts^* control is needed for Figure 2—figure supplement 1.

5) G-trace with low- and high-flp needs to be better described in the text/(Figure 3—figure supplement 1).

6) Figure 8A-H, MMP1-lacZ is robustly upregulated in CtBP homozygous mutant clones, whereas, AP-1-GFP is only found in a few cells of CtBP mutant clones. What causes the dramatic difference of these two JNK signaling reporters upon CtBP loss?

Reviewer #3:

The manuscript by Worley and colleagues seeks to understand mechanisms of cellular reprogramming during tissue regeneration while keeping in check inappropriate cell fates and proliferation. The authors use the *Drosophila* wing disc as a model of regeneration after induced cell death in order to screen for factors that modify the formation of regenerated wings and wing tissue. This genetics screen reveals that the C-terminal Binding Protein (CtBP) likely inhibits changes in cell fates and limits inappropriate differentiation and proliferation. Through a series of tenacious and elegant genetic experiments the authors show that a combination of JAK/STAT activation and AP-1 activity cell specifically located at the "weak point" of the wing disc notum can switch cell fates when triggered to regenerate after tissue damage. Reduction of CtBP function (either in a heterozygous background or homozygous clones) increases AP-1, as assayed by an AP-1-GFP reporter and is proposed to increase the levels of cell fate changes. Although it is not clear how CtBP modulates AP-1 levels and/or what other CtBP target genes contribute to restricting cell fate changes, these observations are interesting in and of themselves as they lay the foundation upon which more molecular mechanistic experiments can be done. For example, what exactly constitutes "weak points", how does CtBP co-repressor work to modulate AP-1 activity specifically at the proposed weak point, and consequently are Upd-family ligands directly involved in cell fate determination in the context of regenerating tissue? Although this study identifies CtBP as a previously unrecognized modulator of cellular plasticity, it remains unclear how it achieves this and how much more molecular mechanism could/should have been elucidated. Other than that I do not have substantive concerns.

---

## [Author Response]

Reviewing Editor's comments:All three reviewers appreciated the approaches used in this paper, and its content. However reviewer #1 felt that the paper was much too long and had many inconclusive sections that could be condensed or removed. In discussions, the other reviewers agreed with this perspective.Reviewer #3 also felt that the paper should have delivered more mechanistic insight as to how CtBP affects transdetermination, but we understand that providing this could be challenging and time-consuming, and so we decided that it is not required at this stage. Hence we would like to invite you to submit a revised manuscript that is significantly shorter than the current version (30-50%) and more linear in presentation. Specific suggestions for how to do this can be found in the reviews, which I include below essentially in their entirety.

Thank you for the opportunity to revise the manuscript. We have rewritten the manuscript completely. The Introduction and Results and Discussion section is now 30% shorter. Importantly we feel that the logical flow is much improved. We took three months (rather than the two months you suggested) because some of the additional experiments took a bit longer (One generation in our ablation/regeneration system takes 22-25 days).

In addition, the points #4 and #6 from reviewer 1 merit special consideration – if you can provide further data addressing these points this would be much appreciated. At the very least, a clear discussion of the action of egr is needed. The reviewers also had a number of minor comments regarding controls, clarification of methods and interpretation of the results, which you should address in your revision. Overall we agreed that the paper was sufficiently developed and could be published without large amounts of additional data, provided the delivery is succinct and clear. eLife looks forward to receiving your revision.

We have included additional data on the action of *eiger* (as detailed in responses below to specific reviewer comments).

Reviewer #1:In this paper, Worley, Alexander, and Hariharan revisit the phenomenon of damage-initiated trans-determination in Drosophila wing imaginal discs, using new genetic methods to identify the genes involved. This is an interesting process with potentially broad relevance, but little progress has been made on it in years. The authors present a very nice genetic saturation screen of the third chromosome (~40% of the genome) that identifies many genes potentially involved. Then they focus on one gene, CtBP at 87D, and show how this affects transdetermination of notum cells to wing pouch cells. The process involves JNK and STAT activation (as expected) at the wound site, is multi clonal, and involves the activation of a number of other genes that are tracked. The effects of these genes are are clearly demonstrated, but the paper falls short of determining any clear mechanisms at the molecular level.This manuscript starts out nicely with one of the best introductions on the subject of transdetermination that I've read, and succinct descriptions of the damage protocol (already detailed in a previous publication) and the modifier screen, which is clever and effective (Figure 1). The effect of CtBP is clear, but it was not obvious why the authors focused on this one gene, and this should be stated. Following this nice entry, the paper details a huge number of genetic tests of mechanism in the subsequent 7 figures and 15 pages of meandering text. The specific tests that were done are all logical, and the results are interpreted in a critical fashion, but unfortunately they don't come together in a very readable, coherent form. Although the effects of egr, Ap1, Stat, etc are clear, the authors don't succeed in demonstrating how these factors work together very well, and some obvious questions are left unresolved (see below). In addition, many tests that were relatively inconclusive are presented, as well as quite a few negative results and dead ends. Many paragraphs in the Results section don't have a conclusion at all. Overall the Results section meanders from one issue to the next, with too much discussion, and no obvious direction. It is far too long. It seems to be a very raw draft of a large body of work, essentially un-refined, and it really should be re-written in a much more concise, linear format to be digestible by a general audience. Much of what is here could probably be presented much more simply, or just omitted, without affecting the paper's final conclusions. I suggest trying to pare it down to about half the current length. While I think the core content of this paper is valuable and should be published, I cannot recommend it in its current form. Specific points follow below:

We have put a lot of effort in re-writing the manuscript. It is much shorter, and we think it now has a clear logical flow. We have removed sections that add little to our final conclusions.

1) There is a long section about the Rga locus but then this locus is never mentioned again. What is it's map location? What is it's relevance to the paper's overall conclusion? This locus justifies the screen, but if it is not relevant to the paper's conclusions the section devoted to it should be removed. The paper contains several other examples of discussion of this sort of irrelevant, distracting data.

We have removed the section on the *Rga* locus since it is not particularly relevant to the overall conclusions of our study.

2) The authors should note why they focused on region 87D after doing the screen. Further discussion of other loci that strongly affected transdetermination could enhance this paper, if the genes involved are known. Nevertheless, it is very nice that the authors include the complete results of their very elegant screen (Figure 1, Supplementary file 1).

The reason we focused on 87D was because this is the only region thus far, where we have been able to unambiguously identify the gene within the deletion, that when mutated, robustly enhanced the frequency of notum-to-pouch fate change. We did include all the data from our deficiency screen so others could use our data to explore candidate genes if they wish.

3) The long paragraph in subsection “Re-specifying cell fates requires the activation of the JNK/AP-1 pathway” is an example of a discussion of results that meanders and has no conclusion.

We have eliminated this and other similar paragraphs.

4) There are quite a few experiments that address the requirements of egr, Jnk, and apoptosis in promoting transdetermination, and much discussion about this. However the paper doesn't seem to have a clear conclusion about whether egr triggers transdetermination via JNK-mediated AP1 activation, or JNK-mediated apoptosis, or both. Egr is clearly sufficient, and JNK required but not sufficient, and apoptosis is not sufficient, but it's not clear what, downstream of egr, is doing it. Is apoptosis required? I think a few more experiments are required here, and a cleaner more definitive conclusion is needed about which downstream targets of egr are critical.

We have included additional data that address these issues. Our experiments show that apoptosis of the pouch, in itself does not promote the notum-to-pouch transformation. Ablation with *reaper* does not cause ectopic pouch formation nor does hastening death by co-expressing *eiger* and *reaper*. Our data suggest that cells with high levels of AP-1 activation, express target genes such as *unpaired*-family genes and *dilp8* which contribute to the fate change. We also show that strictly cell autonomous activation of Eiger signaling in the pouch does not suffice. It is possible that this is because cells in the hinge are more capable of expressing high levels of Upd-family proteins than cells in the pouch and do so when they are exposed to Eiger.

5) The effects on dILP8 expression are interesting and suggestive (“dilp8 expression in a secondary location is predictive of ectopic wing pouch formation”), but dILP8 function is not tested. It would be a nice addition to the paper if the role of dILP8 were tested using mutants, RNAi, and overexpression.

Thank you for this suggestion. We have included additional data that show that reducing *dilp8* function reduces the frequency of ectopic pouches and increasing *dilp8* expression increases the frequency.

6) Upd ligand production, in response to egr, or Ap1, or apoptosis directly, seems to be an important aspect of the authors proposed mechanism (Figure 8Q, and Results). However they don't seem to have tested the requirement functionally, except by Upd1 overexpression and in Stat+/- mutants. There are upd2, upd3, and upd2,3 deletion lines that are viable and could be used to test the requirement of these damage-dependent ligands.

We have now included these experiments. The *upd 2,3* deletion dominantly reduces the frequency of ectopic pouches and this effect is overcome by co-expression of *upd1*. These experiments show that *upd 2* and *3* are necessary for ectopic pouch formation and that the different upd family members can function interchangeably in this situation. This strengthens our argument that signaling from the damaged wing pouch to the notum is important for triggering this fate change.

Reviewer #2:[…]Specific Comments:1) Figure 4. Since temperature shift and genetic background affect the frequency of ectopic wings, in order to conclude that "expression of >egr in the wing pouch is the major cause of generating ectopic wings", the authors also need to test the R15B03-, R76A01-, and dpp-Gal4 drivers using the same temperature shift protocol.

We conducted these experiments with *tub-GAL80^ts^* and the same temperature shift protocol. We did not observe ectopic wings. The data are shown in (Figure 5—figure supplement 3).

2) Scale bar is missing in some figures.

We have added scale bars to all of the main figures and to most of the figure supplements.

3) Since the clean genetic background did not produce ectopic wings in rn^ts^>egr, the description at the beginning part of the manuscript is misleading. This information should be included in the Abstract.

We have changed the relevant sentence in the Abstract to be more specific:

“As a result of this screen, we found that reducing function of the transcriptional co-repressor C-terminal Binding Protein (CtBP) results in an increased frequency of inappropriate changes in cell fate.”

has been changed to

“As a result of this screen, we found that reducing function of the transcriptional co-repressor C-terminal Binding Protein (CtBP) results in inappropriate changes in cell fate, which are not observed in wild type.”

4) rn^ts^ control is needed for Figure 2—figure supplement 1.

We have added the non-ablated controls in Figure 2 and Figure 2—figure supplement 1.

5) G-trace with low- and high-flp needs to be better described in the text/(Figure 3—figure supplement 1).

We have re-written this portion of the text to be clearer.

6) Figure 8A-H, MMP1-lacZ is robustly upregulated in CtBP homozygous mutant clones, whereas, AP-1-GFP is only found in a few cells of CtBP mutant clones. What causes the dramatic difference of these two JNK signaling reporters upon CtBP loss?

We have consistently observed a “boundary effect” with the *AP-1-GFP* reporter, which we also observe with *dilp8-GFP.* We have no good explanation for this phenomenon. The *MMP1-lacZ* reporter appears to have a basal level of expression levels throughout the disc, and may also contain an enhancer that is more responsive to the loss of *CtBP*. Note that we do not detect MMP1 protein in *CtBP-/-* mutant clones and therefore the *MMP1-lacZ* reporter is active but not the endogenous gene. In addition, *MMP1-lacZ* reporter expression was detected with an antibody, which may detect β-GAL produced over a longer time window, while both *AP-1-GFP* and *dilp8-GFP* were observed without an antibody. This could account for some of the differences.